


# Bayesian method for estimating Weibull parameters for wind resource assessment in the Equatorial region: a comparison between two-parameter and three-parameter Weibull distributions

M. Golam Mustafa Khan, M. Rafiuddin Ahmed

School of Information Technology, Engineering, Mathematics and Physics, The University of the South Pacific, Laucala Campus, Suva, Republic of Fiji

*Correspondence to*: M. Rafiuddin Ahmed (ahmed_r@usp.ac.fj)

**Abstract.** The two-parameter Weibull distribution has garnered much attention in the assessment of wind energy potential.
The estimation of the shape and scale parameters of the distribution has brought forth a successful tool for the wind energy industry. However, it may be inappropriate to use the two-parameter Weibull distribution to assess energy at every location, especially at sites where low wind speeds are frequent, such as the Equatorial region. In this work, a robust technique for wind resource assessment using a Bayesian approach for estimating Weibull parameters is first proposed. Secondly, the wind resource assessment techniques using a two-parameter Weibull distribution and a three-parameter Weibull distribution, which
is a generalized form of two-parameter Weibull distribution, are compared. Simulation studies confirm that the Bayesian approach seems a more robust technique for accurate estimation of Weibull parameters. The research is conducted using data from seven sites in Equatorial region from 1º N of Equator to 21º South of Equator. Results reveal that a three-parameter Weibull distribution with non-zero shift parameter is a better fit for wind data having a higher percentage of low wind speeds (0-1 m/s) and low skewness. However, wind data with a smaller percentage of low wind speeds and high skewness showed
better results with a two-parameter distribution that is a special case of three-parameter Weibull distribution with zero shift parameter. The results also demonstrate that the proposed Bayesian approach and application of a three-parameter Weibull distribution are extremely useful for accurate estimate of wind power and annual energy production.

**Keywords:** Wind energy resource assessment; Two-parameter Weibull distribution; Three-parameter Weibull distribution;
Bayesian estimation method.

## 1 Introduction

Wind energy has now become one of the world's fastest-growing sources of energy. It is an inexhaustible source of energy with increasing utilization all around the world. Growing climate change concerns have prompted many developed and developing countries to implement policies that reduce their reliance on non-renewable sources and instead utilize renewable
sources such as wind, hydro, and solar energies (Mostafaeipour et al., 2014). However, developing countries encounter several





challenges in generating sustainable wind energy. There is a need for reliable wind data and proper assessments of a country's wind energy potential before initiating energy generation projects that would help them meet the sustainable development goals set by the United Nations.

While climate change is being experienced globally, some regions are getting affected more than the others. Pacific islands countries (PICs), particularly those in the warmer Equatorial region, are more susceptible to its effects. The contribution of the PICs to the current global greenhouse gas emissions is below 0.03%; yet they are among the first to be affected. It is projected that the people of PICs will be among the first who will need to adapt to climate change or be required to relocate from or abandon their traditional homeland. Some islands are already facing the impacts of climate change on their communities, infrastructure, water supply, coastal and forest ecosystems, fisheries, agriculture, and human health. Island states such as Kiribati, Marshall Islands, Tokelau and Tuvalu are the immediate victims of this phenomenon due to rising sea level. Knowledge of the effects of climate change on PICs should act as a driving force behind the commitment to decrease greenhouse emissions. The PICs, which currently depend heavily on imported fossil fuels and their by-products, need to become more energy efficient and self-reliant (Mohanty, 2012). PICs are among the countries with lowest access to electricity and the prices of electricity are among the highest in the world due to their heavy reliance on high-cost diesel-based generation. Energy security and low-cost energy are becoming increasingly important within the region, which requires increasing investments in renewable energy technologies. PICs are also among the most vulnerable countries to natural disasters. The energy sector can be highly vulnerable to such events, which requires adequate attention to these issues in the design of energy production and distribution infrastructure. This can only be achieved by adopting resilient renewable energy policies. Most of the countries in the region have their national sustainable development plans to achieve United Nations' sustainable development goals (SDGs); for examples, the country Cook Islands, aims to have 100% renewable power generation in near future and Fiji is committed to reducing 30% of its national greenhouse emissions and achieve 99% renewable energy generation by 2030 (MOE, 2017).

However, lack of reliable and accurate wind resource data acts as a barrier to a clean energy future in the PICs, especially in the smaller developing islands (Kidmo et al., 2015). So far, wind resource assessment has received only limited attention in the PICs, and there is a need for further wind data collection and analysis and accurate wind energy potential assessment. World Bank provides support to PICs through the Sustainable Energy Industry Development Project (SEIDP). In various phases of renewable energy resource mapping, they support the countries to carry out an assessment of solar and wind potential. The objectives of this component are to enhance awareness and knowledge of the potential for renewable technologies (solar and wind) to the governments, power utilities and private sector, and to provide governments with a spatial planning framework to guide investments in the renewable energy sector (PPA, 2015).

The utilization of wind energy is slowly increasing in PICs such as New Caledonia, Fiji, Vanuatu, Cook Islands and Samoa with the installation of wind farms. However, there have been little to no attempts to establish wind power in many of these



countries. The University of the South Pacific installed towers of 34 m height, named as Integrated Renewable Energy Resource Assessment Systems (IRERAS), in Kiribati, Nauru, Niue, Tuvalu, Tokelau, Samoa, Tonga, Fiji, Vanuatu, Solomon
and Cook Islands to collect data on wind and solar energy resources (Gosai, 2014).

The Weibull distribution has now become a widely accepted model in determining the potential of wind energy (Indhumathy et al., 2014). Wind energy professionals in different parts of the world have widely employed the use of Weibull distributions in the statistical analyses of wind characteristics and for estimating wind power density (Corotis et al., 1978). The Weibull shape parameter defines the width of wind distribution. A higher shape parameter indicates that the distribution is narrower,
and the peak value is higher. The Weibull scale parameter controls the abscissa scale of the data distribution plot (Chang, 2011). Thus, the Weibull distribution function is comprehensively used for analyzing the wind power potential at a site.

Past researchers found the two-parameter Weibull distribution to be a useful and practical tool for wind energy estimation. The advantages of two-parameter Weibull distribution include its flexibility, simplicity in parameter estimation, convenience of conducting goodness-of-fit tests on these parameters as well as its dependence only on two parameters that can be expressed
in closed form.

However, some authors suggested that the distribution is not suited for all wind regimes encountered in nature such as regimes with a high percentage of low wind speeds and bimodal distributions. Therefore, its usage cannot be generalized. To minimize errors, a suitable probability density function must be carefully selected for different wind regimes.(Carta et al., 2009; Sukkiramathi and Seshaiah, 2020) Patlakas et al. (2017) emphasized the importance of studying low wind speeds stating that
this information can be included in risk assessment. Leahy and Mckeogh (2013) studied the persistence of low wind speed conditions and their implications on the variability of wind power.

Tuller and Brett (1984) proposed a three-parameter Weibull function for wind data analysis and found that it showed better fitness and flexibility than the two-parameter Weibull function. Recently, some authors utilized the three-parameter Weibull distribution and found that it has more flexibility with improved fitness than the two-parameter Weibull distribution in wind
energy assessments. Wais (2017) compared the two and three-parameter Weibull distribution to find the most appropriate distribution of wind speed. The results revealed that methods other than the three-parameter Weibull distribution cannot account for cases where the frequency of low wind speed is higher. The author compared the wind speeds for three different sites and found that the three-parameter Weibull distribution performed the best when there was a greater frequency of lower wind speeds. Sukkiramathi and Seshaiah (2020) and Wang et al. (2022) also utilized the three-parameter Weibull distribution
for analyzing wind power potential. However, to date, only limited research has been carried out on wind data analysis using the three-parameter Weibull distribution.

Furthermore, many estimation methods have been proposed for estimating Weibull parameters. Among these, maximum likelihood estimation (MLE), a popular frequentist technique, has been widely used for estimating the parameters (Teimouri



and Gupta, 2013). Recently, the Bayesian estimation approach has received a lot of attention from many researchers. Among
them is Ibrahim and Mohammed (2011) who considered the Bayesian survival estimator for Weibull distribution with censored
data. Many authors, including Hossain and Zimmer (2003) and Pandey et al. (2011) did some comparative studies on the
estimation of the Weibull parameters using complete and censored samples, and Lye et al. (1993) determined the Bayes
estimation for the extreme-value reliability function. Guure et al. (2012) examined the performance of MLE and Bayesian
methods for estimating the two-parameter Weibull failure time distribution. However, the use of the Bayesian technique for
modelling wind data and analyzing wind power potential was not explored in their work.

The present work is aimed at comparing the two-parameter and three-parameter Weibull distributions to fit wind speed data
more accurately at seven locations in the Equatorial region, where wind speeds are generally lower. Development of a novel
approach using the Bayesian method for estimating the Weibull parameters is also a part of this work. The results from
Bayesian technique are compared with those of the traditional MLE method to determine a more accurate evaluation method
of wind speed characteristics.

**2 Wind Speed Data Sites**

Wind speed data from seven different sites in the Equatorial region were used in the present work, as shown in Table 1.

**Table 1: Locations of data collection sites.**

| Sites | Location | Country | Measurement period | Topography |
|---|---|---|---|---|
| 1. Tarawa | Latitude 1° 26' N<br>Longitude 173° 00' E | Kiribati | September 2012 to<br>September 2013 | Flat |
| 2. Pentecost | Latitude 15º 41' S<br>Longitude 168º 11' E | Vanuatu | October 2012 to<br>November 2013 | Mountainous<br>terrain |
| 3. Rakiraki | Latitude 17º 22' S<br>Longitude 178º 10' E | Fiji | February 2012 to<br>October 2013 | Flat |
| 4. Kadavu | Latitude 19° 0' S<br>Longitude 178º 15' E | Fiji | January 2018 to<br>December 2018 | Mountainous<br>terrain |
| 5. Rarotonga | Latitude 21° 15' S,<br>Longitude 159° 45' W | Cook<br>Islands | January 2016 to<br>December 2018 | Flat |
| 6. Nuku'alofa | Latitude 21º 15' S<br>Longitude 175º 15' W | Tonga | January 2016 to August<br>2019 | Flat |
| 7. Sanasana | Latitude 18º 6' S<br>Longitude 177° 20' E | Fiji | December 2018 to<br>December 2019 | Flat |



For sites 1, 2 and 3, data were obtained from measurements using 34 m tall towers with the help of sensors described in Table 2. The NRG systems towers, named Integrated Renewable Energy Resource Assessment Systems (IRERAS), with a height of 34 m were used. NRG *SymphoniePlus3* was the data-logger used which was connected to seven different sensors installed on the tower. The sensors measured wind speed, temperature, pressure, rainfall, solar insolation, humidity, and wind direction.

The data were either collected from the SD card in person or sent via the GSM-based network to a data-bank located at the ICT centre of the University of South Pacific at the Laucala Campus, Fiji. The anemometers (serial numbers 179500189054-57, 179500189089-90) have an accuracy of 0.1 m/s and a range of 0.4 to 96 m/s. The wind vane is placed at 30 m above ground level (AGL). The data were recorded in a time-series format in an RWD file which were later transferred to a Microsoft excel sheet. The wind speed data were recorded continuously at an interval of 10 minutes with three cup anemometer – two at heights

of 34 m above ground level and one at 20 m above ground level – respectively. For sites 4, 5 and 6, satellite data were downloaded; land data from ERA5 were used in the present work (Ref: https://cds.climate.copernicus.eu/). ERA5 is the fifth generation ECMWF reanalysis for the global climate and weather for the past 4 to 7 decades. Reanalysis combines model data with observations from across the world into a globally complete and consistent dataset using the laws of physics. This principle, called data assimilation, is based on the method used by numerical weather prediction centres, after certain number

of hours (12 hours at ECMWF) a previous forecast is combined with newly available observations in an optimal way to produce a new best estimate of the state of the atmosphere, called analysis, from which an updated, improved forecast is issued. Reanalysis works in the same way, but at reduced resolution to allow for the provision of a dataset spanning back several decades. Reanalysis does not have the constraint of issuing timely forecasts, so there is more time to collect observations, and when going further back in time, to allow for the ingestion of improved versions of the original observations, which all benefit

the quality of the reanalysis product. For site 7, an NRG systems towers, similar to the ones used for sites 1, 2 and 3, but 50 m high, was installed; it has anemometers at 50 m, 40 m and 30 m AGL.

**Table 2: Specifications of the relevant measurement sensors (Aukitino et al., 2017).**

| Parameter | Sensor Type | Range | Accuracy |
|---|---|---|---|
| **Wind speed** | NRG#40C anemometer | 0.4-96.0 m/s | 0.1 m/s |
| **Wind direction** | NRG 200P direction vane | 0-360º | N/A |
| **Pressure** | NRG BP-20 barometric pressure sensor | 15 kPa - 115 kPa | 1.5 kPa |
| **Temperature** | NRG 110S | -40 ºC – 65 ºC | 1.11 ºC |

For the measured values, some uncertainties were taken into account such as calibration errors, the terrain of the site that was

used, the dynamic over speeding, the error introduced due to wind shear and the inflow angle (Jain, 2016). The measurements in the present work were performed close to the shoreline at a flat terrain. The flow was in the horizontal plane, resulting in a lower uncertainty level. The calibration report for the anemometers used in the present work showed a maximum uncertainty



of 0.6% for a wind speed range of 4-7 m/s, which reduced at higher wind speeds. The overall uncertainty in the estimation of
wind speed is obtained by taking all the above uncertainties into account(Jain, 2016) and using the relation in equation (1):


$$\varepsilon = \sqrt{\sum_{i=1}^{N} \varepsilon_i^2} \tag{1}$$

where $\varepsilon_i$ is each component of uncertainty and $N$ is the number of components of uncertainty. The uncertainties were estimated
at 95% confidence level. As per the IEC Standard IEC 61400-12-1,(Iec, 2017) the uncertainty in the measurements was
estimated to be approximately 1.74%.

**3 Weibull Distribution**

Assessment of wind power energy at a site requires knowledge of the appropriate probability distribution of the site's wind
speed, as the estimation of wind energy depends on its accuracy.

**3.1 Two-parameter Weibull distribution**

The two-parameter Weibull probability density functions (PDF) and the cumulative distribution function (CDF) for wind
speed, U, respectively are given by

$$f(U) = \frac{k}{A}\left(\frac{U}{A}\right)^{k-1} e^{-\left(\frac{U}{A}\right)^k} ; \quad U > 0, k > 0, A > 0 \tag{2}$$

and

$$F(U) = 1 - e^{-\left(\frac{U}{A}\right)^k} \tag{3}$$

where $f(U)$ is the probability of observing the wind speed, $k$ is the shape parameter and $A$ is the scale parameter (m/s) of the
distribution. The parameter $k$ indicates the wind potential and what peak the distribution can reach. Its value ranges between
1 and 3. A lower $k$ value signifies highly variable winds, while constant winds are characterized by a larger $k$. The parameter
$A$ denotes how windy the site under study is and it takes a value proportional to the mean wind speed (Manwell et al., 2010;
Sukkiramathi and Seshaiah, 2020).

**3.2 Two-parameter Weibull distribution**

The three-parameter Weibull PDF and the CDF for wind speed, respectively are given by

$$f(U) = \frac{k}{A}\left(\frac{U-\theta}{A}\right)^{k-1} e^{-\left(\frac{U-\theta}{A}\right)^k} ; \quad U > 0, k > 0, A > 1, -\infty < \theta < \infty \tag{4}$$



and

$$F(U) = 1 - e^{-\left(\frac{U-\theta}{A}\right)^k} \tag{5}$$

where $f(U)$ is the probability of observing the wind speed, k is the shape parameter, A is the scale parameter (m/s), and

θ is the shift or location parameter (m/s) of the distribution. If $\theta = 0$, $f(U)$ and $F(U)$ become the PDF and CDF of a two-parameter Weibull distribution, respectively.

As the name implies, the shift parameter, θ, shifts the distribution along the abscissa. When θ = 0, the distribution starts at U = 0 or at the origin. Whereas, if θ > 0, the distribution starts at the location θ to the right of the origin. If θ < 0, the distribution

starts at the location θ to the left of the origin, although it will not be the case for wind speed. For the distribution of wind speed, θ provides an estimate of the earliest time-to-start the wind (Tuller and Brett, 1984; Wais, 2017).

### 4 Methods of Estimating Weibull Parameters

To estimate the Weibull parameters, we propose a Bayesian approach and compare its performance with a popular frequentist approach, the maximum likelihood estimation (MLE) method.

### 4.1 The Maximum Likelihood Method

#### 4.1.1 Two-parameter distribution

MLE is the most popular technique for deriving estimators.(Aukitino et al., 2017; Casella and Berger, 2020; Chaurasiya et al., 2018) If $U_1,...,U_n$ are the wind speed values with the Weibull density function given in (2), the shape parameter (k) and scale

parameter (A) are the values that maximize the likelihood function $L(k, A|U_1,...,U_n) = \Pi_{i=1}^n f(U_i|k, A)$. Then, solving

$\partial \ln L/\partial k = 0$ and $\partial \ln L/\partial A = 0$ gives the equation of MLE of the scale parameter A as:

$$A = \frac{1}{n} \sum_{i=1}^n U_i^k \tag{6}$$

Finally, equation (7) is used for estimating the shape parameter (k) as:

$$\frac{1}{k} + \frac{1}{n} \sum_{i=1}^n \ln(U_i) - \frac{\sum_{i=1}^n U_i^k \ln(U_i)}{\sum_{i=1}^n U_i^k} = 0 \tag{7}$$

which may be solved to get an estimate of k using Newton-Raphson method or any other numerical procedure because equation

(7) does not have a closed form solution. When the value of k is obtained, the value of A can be found using equation (6).

#### 4.1.2 Three-parameter distribution





The likelihood function $L$ for estimating the parameters is given by $L\left(k, A, \theta \middle| U_1, ..., U_n\right) = \Pi_{i=1}^{n} f\left(U_i \middle| k, A, \theta\right)$. Then, solving $\partial \ln L / \partial k = 0$, $\partial \ln L / \partial A = 0$ and $\partial \ln L / \partial \theta = 0$ gives the equations of MLE of the parameters as shown in Equations (8-10):


$$\frac{n}{k} \sum_{i=1}^{n} \log\left(\frac{U_i - \theta}{A}\right) - \sum_{i=1}^{n} \left(\frac{U_i - \theta}{A}\right)^k \log\left(\frac{U_i - \theta}{A}\right) = 0 \tag{8}$$

$$\frac{nk}{A} + \frac{k}{A} \sum_{i=1}^{n} \left(\frac{U_i - \theta}{A}\right)^k = 0 \tag{9}$$

$$-\left(k-1\right) \sum_{i=1}^{n} \frac{1}{U_i - \theta} - \frac{k}{A} \sum_{i=1}^{n} \left(\frac{U_i - \theta}{A}\right)^{k-1} = 0 \tag{10}$$

There is no closed form solution of the equations but the non-linear equations (8) - (10) may be solved by applying some optimization techniques such as Newton-Raphson method or other numerical procedures (Teimouri and Gupta, 2013; Lawless, 195 2003).

### 4.1.3 Evaluation of MLE methods

To determine the best model, we can compare the fit of the two MLE methods using different measures of goodness-of-fit (Luceño, 2008; Cousineau and Allan, 2015; Ramachandran and Tsokos, 2021). The most used criteria are:

**Log-likelihood (log-like):**

If a PDF $f_{\hat{\eta}}\left(U\right)$ fitted on the wind speed data and $\hat{\eta}$ is the estimated parameter of the distribution, then the log-likelihood for the goodness of fit is obtained by the following equation:

$$\text{log-like} = \log\left(\prod_{i=1}^{n} f_{\hat{\eta}}\left(U_i\right)\right) \tag{11}$$

where $U_i$ is the $i$th observed wind speed and $n$ is the number of observations in the dataset. A higher value of log-likelihood value indicates a better fit.

**Akaike information criteria (AIC):**

If $k$ is the number of distribution parameters to estimate, AIC is obtained using equation (12):

$$\text{AIC} = -2\left(\text{log-like}\right) + 2k \tag{12}$$

A lower value of AIC indicates that the model fits the data better. Compared to the log-likelihood, this criterion takes into consideration the parsimony of the model as it includes a penalty term that increases the number of parameters.

**Bayesian information criteria (BIC):**



This criterion is obtained using equation (13):

$$\text{BIC} = -2\left(\text{log-like}\right) + k\,\log(n) \tag{13}$$

Similar to AIC, a lower value of BIC indicates that the model fits the data better. However, BIC provides a stronger penalty than AIC for additional parameters.

**Kolmogorov-Smirnov (KS) test:**

The Kolmogorov-Smirnov (KS) test is also used to check the adequacy of a given theoretical distribution for a given set of wind speed data. The KS test computes the maximum difference between the predicted and observed distribution, and the test statistic $D$ is given by:

$$\text{D} = \max_{1 \le i \le n}\left|F_i - \hat{F}_i\right| \tag{14}$$

where $\hat{F}_i$ is the $i$th predicted cumulative probability from the theoretical CDF and $F_i$ is the empirical probability of the $i$th observed wind speed.

**Anderson-Darling (AD) test:**

For a finite data sample, the Anderson-Darling (AD) test statistic $A^2$ is defined by:

$$A^2 = -n - s \, , \tag{15}$$

where $s = \sum_{i=1}^{n} \dfrac{2i-1}{n}\left[\log\left(\hat{F}_i\right) + \log\left(1 - \hat{F}_{n-i+1}\right)\right]$.

**4.2 The Bayesian Method**

A classical frequentist approach such as MLE has certain drawbacks. Most of its properties hold only for large sample sizes and it requires a symmetric form of sample distribution. The Bayesian approach, however, is free from such limitations. Moreover, Bayesian simulation tools provide an exact method of inference even if sample size is very small. Thus, in real-life situations where the sample size may be small, Bayesian methods seem to be more suitable over frequentist methods if prior information about the parameters is available.

In this paper, a Bayesian inference approach for modeling of wind speed data is proposed. In the Bayesian paradigm, data and prior information about the parameters are combined together to make an inference about the parameters of interest.

The most influential contribution of the Bayesian approach is its modification of the likelihood function into a posterior - a valid probability distribution defined by the classic Bayes' rule. The posterior distribution of wind speed is expressed as:



$$p(\eta|U) = \frac{p(U|\eta)\,p(\eta)}{p(U)} \tag{16}$$

where $p(\eta|U)$ is the posterior distribution of wind speed, $p(\eta)$ is the prior distribution of unknown parameters $\eta = (k, A, \theta)$, $p(U|\eta)$ is the likelihood of wind speed data and $p(U) = \int p(U|\eta)\,p(\eta)\,d\eta$. The denominator of equation (16), $p(U)$, normalizes the posterior distribution, $p(\eta|U)$. Since it is independent of $U$, it is often convenient to write the posterior distribution as:

$$p(\eta|U) \propto p(U|\eta)\,p(\eta) \tag{17}$$

i.e. the posterior distribution of the parameters is proportional to the likelihood function times the prior distribution of parameters. While fitting wind speed data, a non-informative uniform prior distribution is used, as very little prior knowledge about its model parameters is available. In Bayesian computations, a sample of the joint posterior distribution is obtained by using Gibbs sampler to simulate a sample from a Markov Chain Monte Carlo (MCMC). Then, we can calculate the desired values of the posterior.

In this paper, the software JAGS is used to fit the model. The R package R2jags is used to summarize the posterior inference, which is discussed in more detail in Section 4.2.1.

### 4.2.1   Bayesian Fitting of Weibull distribution with JAGS

JAGS, an acronyms for "Just Another Gibbs Sampler" (Plummer, 2003), accepts a model string written in an R-like syntax
that compiles and generates MCMC samples from the model using Gibbs sampling. It is an open-source software written in C++ using GNU compilers and packaging tools - freely available at http://mcmc-jags.sourceforge.net/. R packages such as R2jags or rjags allow running JAGS models within R on Windows machines for the summarization of posterior inference.

In the present work, JAGS models for the Bayesian fit of wind speed with two and three parameter Weibull distributions are
developed and MCMC data are generated. In MCMC simulations, the Gibbs sampler with the JAGS models is run for 10,000 iterations using the jags function in R2jags package (Su and Yajima, 2020). Then, the posterior estimates of parameters are obtained by performing the Gibbs sampler iterations and using 1000 burn-in period to attain convergence with five thinning intervals and 3 chains with a sample size of 1800 per chain. The model specifications to perform the Bayesian fit, and the jags function and its arguments are presented in Appendix 1.

### 4.2.2    Evaluation of Bayesian Models





The standard likelihood, AIC and BIC statistics, discussed in Section 4.1.3 are not relevant while evaluating Bayesian methods such as MCMC. Instead, Spiegelhalter et al. (2002) suggested that the Deviance Information Criterion (DIC) should be used to compare models. The DIC is a generalization of AIC that is based on Deviance statistics:

$$D(\eta) = -2 \log f(U/\eta) + 2 \log h(U) \tag{18}$$

where $h(U)$ is some standardizing function of the data. The DIC is then defined as:

$$DIC = \bar{D} + p_D \tag{19}$$

where $\bar{D} = E_{\eta|U}(D)$ is the posterior expectation of deviance and $p_D$ is effective number of parameters that captures the complexity of a model. A smaller value of DIC indicates a better-fitting model.

**5 Estimate of wind power and energy**

When the wind speed (U) of a site and the frequency distribution $f(U)$ are known, the available wind power and wind energy can be estimated. If $\rho$ is the density of air, $D$ is the turbine rotor diameter and $A_R = \pi D^2/4$ is the rotor cross sectional area, then the probability of available wind power for a given velocity $U$ is obtained by

$$p(U) = \frac{1}{2} \rho A_R U^3; \quad U > 0 \tag{20}$$

Then, the expected wind power ($P$) is estimated by

$$\text{Wind power, } P = \int_0^\infty P(U).f(U)\,dU \tag{21}$$

Substituting (20), (2) and (4) in (21), the expected wind powers for two-parameter and three-parameter Weibull distributions respectively, are determined by

$$P_{2P} = \frac{1}{2} \rho A_R \frac{k}{A^k} \int_0^\infty U^{k+2} e^{-\left(\frac{U}{A}\right)^k} dU \tag{22}$$

and

$$P_{3P} = \frac{1}{2} \rho A_R \frac{k}{A^k} \int_0^\infty U^3.(U-\theta)^{k-1} e^{-\left(\frac{U-\theta}{A}\right)^k} dU \tag{23}$$

If $t_{Year} = 8760$ is the total number of hours in a year, the expected wind energy for two-parameter and three-parameter Weibull distributions respectively are determined by

$$E_{2P} = P_{2P} \cdot t_{Year} \tag{24}$$

and

$$E_{3P} = P_{3P} \cdot t_{Year} \tag{25}$$





## 6 Analyzing performance of different estimators

The efficiency and performance of MLE and Bayesian methods for estimating two and three parameter Weibull distributions

were determined using different goodness of fit and error measures such as coefficient of determination ($R^2$), root mean square

error (RMSE), coefficient of efficiency (COE), mean absolute error (MAE) and the mean absolute percentage error (MAPE).

Arithmetically, these are computed as follows (Kidmo et al., 2015; Aukitino et al., 2017; Azad et al., 2014):

### Coefficient of determination ($R^2$):

It is a statistical measure that gives some information about the goodness of fit of a model, that is, how much the variance of

the observed data is explained by the fitted model. It is defined as:


$$R^2 = 1 - \frac{\sum_{i=1}^{n}(U_i - \hat{U}_i)^2}{\sum_{i=1}^{n}(U_i - \bar{U})^2} \tag{26}$$

where $n$ is the number of observations, $U_i$ is the $i$th actual data, $\hat{U}_i$ is the $i$th predicted data with the Weibull distribution, $\bar{U}$

is the mean of actual data. A higher $R^2$ value indicates a better fit and $R^2 = 1$ indicates that the regression predictions perfectly

fit the data.

### Root mean square error (RMSE):

It determines the deviation of the predicted values of wind speed from the observed values and obtained by

$$RMSE = \left[ \frac{1}{n} \sum_{i=1}^{n} \left( U_i - \hat{U}_i \right)^2 \right]^{\frac{1}{2}} \tag{27}$$

A smaller RMSE value normally indicates accurate modeling. The calculated RMSE value approaches zero as the difference

between the observed and predicted values becomes smaller (Indhumathy et al., 2014).

### Coefficient of efficiency (COE):

It quantifies the ratio of difference between predicted wind speed and the mean wind speed to the difference between actual

values and the average of wind speeds. A higher COE value indicates a good fit for the data. It is expressed as:

$$COE = \frac{\sum_{i=1}^{n} \left( \hat{U}_i - \bar{U} \right)^2}{\sum_{i=1}^{n} \left( U_i - \bar{U} \right)^2} \tag{28}$$

### Mean absolute error (MAE):

The mean absolute error is a measure of the absolute difference between predicted and actual values. A smaller value of

MAE indicates higher accuracy. The MAE is mathematically expressed as:



$$MAE = \frac{1}{n}\sum_{i=1}^{n}\left|\hat{U}_i - U_i\right| \qquad (29)$$

**Mean absolute percentage error (MAPE):**

It is a comparative measure, indicating the error as a percentage of the actual data which helps accurately predict the forecasting method. Like MAE, a lower value of MAPE indicates better accuracy. It is mathematically expressed as:


$$MAPE = \frac{100}{n}\sum_{i=1}^{n}\left|\frac{\hat{U}_i - U_i}{U_i}\right| \qquad (30)$$

## 7 Results

In this section, the results of fitting of two-parameter (2-p) and three-parameter (3-p) Weibull distributions are presented. Further, the results for the application of MLE and the proposed Bayesian approach for estimating the parameters, as described in Sections 3 and 4, are also presented. To accomplish this, wind speed data at seven different sites were collected as mentioned

in Section 2. Table 3 provides wind speed distributions at these sites. The table shows that the range of speed varies at different sites. The lowest range of wind speed was observed at site 1 (0-19 m/s) and the highest range was found at site 3 (0-34 m/s). Some sites tend to have more low to null speed (0-1 m/s).

**Table 3: Frequency distributions of wind speed at different sites.**

| Wind speed $U$ | Site 1 Frequency | Site 2 Frequency | Site 3 Frequency | Site 4 Frequency | Site 5 Frequency | Site 6 Frequency | Site 7 Frequency |
|---|---|---|---|---|---|---|---|
| (0,1] | 889 | 1063 | 5882 | 62 | 209 | 251 | 1662 |
| (1,2] | 2197 | 1998 | 4246 | 219 | 773 | 799 | 2161 |
| (2,3] | 5224 | 3579 | 4951 | 367 | 1498 | 1543 | 3712 |
| (3,4] | 7006 | 5739 | 5845 | 641 | 1928 | 2390 | 5220 |
| (4,5] | 8721 | 8077 | 7134 | 803 | 2388 | 3330 | 6331 |
| (5,6] | 8783 | 10332 | 8716 | 949 | 2878 | 4260 | 7393 |
| (6,7] | 7554 | 9570 | 10771 | 1026 | 2965 | 4590 | 6698 |
| (7,8] | 5591 | 7094 | 11215 | 1052 | 2937 | 3915 | 5651 |
| (8,9] | 3381 | 4810 | 9363 | 851 | 2833 | 3502 | 4449 |
| (9,10] | 1706 | 2701 | 6496 | 882 | 2349 | 2812 | 3535 |
| (10,11] | 604 | 1255 | 4226 | 723 | 2016 | 2023 | 2792 |
| (11,12] | 236 | 482 | 2366 | 437 | 1411 | 1205 | 1989 |
| (12,13] | 125 | 119 | 1177 | 311 | 892 | 715 | 1316 |
| (13,14] | 42 | 26 | 588 | 215 | 661 | 278 | 654 |
| (14,15] | 11 | 9 | 247 | 99 | 405 | 193 | 288 |
| (15,16] | 3 | 8 | 107 | 54 | 91 | 114 | 67 |
| (16,17] | 1 | 10 | 60 | 7 | 42 | 97 | 9 |
| (17,18] | 4 | 3 | 53 | 9 | 17 | 43 | 1 |
| (18,19] | 2 | 3 | 38 | 6 | 6 | 30 | 0 |
| (19,20] | 0 | 1 | 13 | 6 | 2 | 16 | 0 |
| (20,21] | 0 | 0 | 5 | 7 | 3 | 9 | 0 |
| (21,22] | 0 | 0 | 4 | 15 | 0 | 8 | 0 |





| | | | | | | | |
|---|---|---|---|---|---|---|---|
| (22,23] | 0 | 0 | 3 | 8 | 0 | 11 | 0 |
| (23,24] | 0 | 0 | 6 | 2 | 0 | 1 | 0 |
| (24,25] | 0 | 0 | 5 | 5 | 0 | 1 | 0 |
| (25,26] | 0 | 0 | 4 | 2 | 0 | 0 | 0 |
| (26,27] | 0 | 0 | 4 | 0 | 0 | 0 | 0 |
| (27,28] | 0 | 0 | 3 | 1 | 0 | 0 | 0 |
| (28,29] | 0 | 0 | 5 | 0 | 0 | 0 | 0 |
| (29,30] | 0 | 0 | 10 | 0 | 0 | 0 | 0 |
| Above 30 | 0 | 0 | 4 | 1 | 0 | 0 | 0 |
| Total frequency | 52080 | 56879 | 83554 | 8760 | 26304 | 32136 | 53928 |


Both the 2-p and 3-p Weibull distributions were fitted to the recorded wind speed data and the parameters in the distributions were estimated using the MLE and the Bayesian methods. In the MLE method, the goodness of fit with 2-p and 3-p Weibull distributions is evaluated using the statistical measures AIC, BIC, AD, KS and log-like. Table 4 presents the estimated values of the parameters of 2-p and 3-p Weibull distributions and the values of the various statistical measures determined by the

MLE at the seven sites.

**Table 4: MLE estimated values of parameters and statistical measures.**

| Site | Distribution | $k$ | A | $\theta$ | AIC | BIC | AD | KS | log-like |
|---|---|---|---|---|---|---|---|---|---|
| 1 | 2-p Weibull | 2.564486 | 6.019568 | | 230352.8 | 230370.6 | 23.95999 | 0.021077 | -115174.4 |
| | 3-p Weibull | 2.777792 | 6.438856 | -0.380611 | **230075.7** | **230102.3** | **9.98482** | **0.013953** | **-115034.9** |
| 2 | 2-p Weibull | 2.735978 | 6.535565 | | 256112.4 | 256130.3 | 101.4772 | 0.038265 | -128054.2 |
| | 3-p Weibull | 3.233794 | 7.532297 | -0.922046 | **255252.9** | **255279.8** | **32.0876** | **0.025796** | **-127623.5** |
| 3 | 2-p Weibull | 1.948074 | 7.017269 | | 433669.2 | 433687.8 | 1414.721 | 0.092759 | -216832.6 |
| | 3-p Weibull | 2.636074 | 8.804806 | -1.546749 | **429666.9** | **429694.9** | **478.748** | **0.058134** | **-214830.5** |
| 4 | 2-p Weibull | 2.385540 | 8.445309 | | **45434.68** | **45448.83** | 6.759304 | 0.019735 | -22715.34 |
| | 3-p Weibull | 2.401323 | 8.493057 | -0.042536 | 45435.82 | 45457.05 | 6.577964 | 0.019739 | **-22714.91** |
| 5 | 2-p Weibull | 2.438715 | 8.234657 | | 135020.5 | 135036.9 | 16.29118 | 0.015911 | -67508.26 |
| | 3-p Weibull | 2.569510 | 8.596043 | -0.323893 | **134971.8** | **134996.4** | **10.37910** | **0.014681** | **-67482.92** |
| 6 | 2-p Weibull | 2.503177 | 7.843472 | | 159616.8 | **159633.6** | 22.04708 | 0.019279 | -79806.41 |
| | 3-p Weibull | 2.522584 | 7.896873 | -0.047855 | **159615.8** | 159641.0 | **21.85474** | **0.018862** | **-79804.91** |
| 7 | 2-p Weibull | 2.153354 | 7.112349 | | 271584.1 | 271601.9 | 69.32922 | 0.021085 | -135790.1 |
| | 3-p Weibull | 2.425388 | 7.807676 | -0.603869 | **270987.3** | **271014.0** | **20.57445** | **0.010117** | **-135490.6** |

In Bayesian estimates, uniform prior distributions of the parameters were used to fit wind data. Firstly, a sample of the joint posterior distribution by simulating a sample from MCMC methods using a Gibbs sampler as discussed in Section 4.2.1 is

obtained. Finally, the DIC is obtained to evaluate the Bayesian parameters of the Weibull distributions. Table 5 presents the estimated mean values of the parameters with standard deviation (SD) of both two and three parameter Weibull distributions for all the sites. The 95% credible region (lower limit - 2.5% and upper limit - 97.5%) for each of the parameters and the model evaluation statistic DIC values are also presented.

**Table 5: Estimated values of the parameters obtained using Bayesian technique and summary statistics.**

| Site | Distribution | Parameter | Mean | SD | 2.5% | 97.5% | DIC |
|---|---|---|---|---|---|---|---|
| 1 | 2-p Weibull | k | 2.564487 | 0.008762 | 2.547292 | 2.581560 | 949864.6 |



| Site | Model | Param | Mean | SD | 2.5% | 97.5% | DIC |
|------|-------|-------|------|-----|------|-------|-----|
| | | A | 6.019417 | 0.010876 | 5.998086 | 6.041268 | |
| | 3-p Weibull | k | 2.778195 | 0.018610 | 2.742305 | 2.814923 | **949587.6** |
| | | A | 6.439670 | 0.033898 | 6.373026 | 6.507357 | |
| | | $\theta$ | -0.381144 | 0.029685 | **-0.439400** | **-0.323193** | |
| 2 | 2 parameter | k | 2.736076 | 0.008940 | 2.718410 | 2.753285 | 256112.2 |
| | | A | 6.535458 | 0.010427 | 6.515002 | 6.555774 | |
| | 3 parameter | k | 3.231943 | 0.023096 | 3.184961 | 3.273617 | **255252.4** |
| | | A | 7.528328 | 0.044253 | 7.438378 | 7.607035 | |
| | | $\theta$ | -0.918508 | 0.040458 | **-0.989749** | **-0.835880** | |
| 3 | 2 parameter | k | 1.947640 | 0.005556 | 1.936812 | 1.958478 | 433669.2 |
| | | A | 7.017671 | 0.013022 | 6.992180 | 7.042895 | |
| | 3 parameter | k | 2.635850 | 0.016118 | 2.604492 | 2.667974 | **429666.8** |
| | | A | 8.804653 | 0.044092 | 8.718409 | 8.892835 | |
| | | $\theta$ | -1.546999 | 0.038766 | **-1.624995** | **-1.472661** | |
| 4 | 2-p Weibull | k | 2.385596 | 0.019486 | 2.347390 | 2.424420 | **166458.7** |
| | | A | 8.446062 | 0.039889 | 8.367275 | 8.525195 | |
| | 3-p Weibull | k | 2.405809 | 0.026990 | 2.356128 | 2.461114 | 166459.8 |
| | | A | 8.507469 | 0.070222 | 8.377540 | 8.660753 | |
| | | $\theta$ | -0.055714 | 0.051466 | **-0.169674** | **0.031573** | |
| 5 | 2-p Weibull | k | 2.438528 | 0.011898 | 2.415595 | 2.462772 | 498423.7 |
| | | A | 8.233447 | 0.021645 | 8.190758 | 8.276122 | |
| | 3-p Weibull | k | 2.569184 | 0.024930 | 2.521912 | 2.619417 | **498375.**1 |
| | | A | 8.596434 | 0.065046 | 8.471752 | 8.729620 | |
| | | $\theta$ | -0.324597 | 0.055076 | **-0.438669** | **-0.220035** | |
| 6 | 2-p Weibull | k | 2.502775 | 0.010741 | 2.482126 | 2.524369 | 603592.3 |
| | | A | 7.843795 | 0.018777 | 7.806656 | 7.880416 | |
| | 3-p Weibull | k | 2.523740 | 0.016243 | 2.492982 | 2.555752 | **603591.3** |
| | | A | 7.900740 | 0.038140 | 7.830501 | 7.978555 | |
| | | $\theta$ | -0.051032 | 0.030016 | **-0.113036** | **0.004279** | |
| 7 | 2-p Weibull | k | 2.153266 | 0.007378 | 2.139086 | 2.167660 | 1016627 |
| | | A | 7.111621 | 0.014636 | 7.083240 | 7.140584 | |
| | 3-p Weibull | k | 2.425984 | 0.016281 | 2.394208 | 2.457741 | **1016030** |
| | | A | 7.808854 | 0.040671 | 7.730066 | 7.892452 | |
| | | $\theta$ | -0.604597 | 0.033817 | **-0.673024** | **-0.539979** | |


The goodness-of-fit criteria and summary statistics presented in Tables 4 and 5 indicate that the **3-p Weibull** distribution fits better than the 2-p Weibull distribution for wind speed at the sites 1, 2, 3, 5 and 7 as all the goodness-of-fit measures (AIC, BIC, AD, KS and log-like) are smaller in MLE estimate and the DIC is also smaller in Bayesian estimate. Moreover, as shown in Table 5 and the marginal posterior means of shift parameter $\theta$ are -0.381144, -0.918508, -1.546999, -0.324597 and -

0.604597. The 95% credible regions of this parameter as shown in columns 2.5% and 97.5% clearly indicates that the **3-p Weibull** distribution is more appropriate at these sites as the value of its shift parameter is non-zero (i.e. $\theta \neq 0$ ).





However, for the site 4, both MLE results in Tables 4 and 5 indicate that the **2-p Weibull** distribution fits better than the 3-p Weibull distribution at this site as the AIC, BIC and KS are smaller in MLE estimate and the DIC is smaller in Bayesian estimate. Although, the marginal posterior mean of $\theta$ of 3-p Weibull distribution is -0.055714 but its 95% credible region (-0.169674, 0.031573) indicates that the value of its shift parameter is zero (i.e. $\theta = 0$). This implies that 3-p Weibull reduces to the 2-p Weibull indicating that **2-p Weibull** distribution is more appropriate at this site.

Whereas, at the site 6, the results show a lack of significant difference between the 2-p and 3-p Weibull distributions while fitting wind speeds, as all MLE and Bayesian estimates are similar in numerical value. Also, the 95% credible region indicates that $\theta = 0$ in Bayesian estimate, which indicates **2-p Weibull** may be a better distribution at this site as the 3-p Weibull distribution reduces to the 2-p Weibull distribution. It should be noted that the 3-p Weibull distribution is a generalized form of 2-p distribution, and when $\theta = 0$, it becomes 2-p distribution. Thus, 2-p distribution can be seen as a special case of 3-p distribution.

Finally, since the Bayesian estimation is a simulation based iterative procedure, the convergence of the model is diagnosed by the visual inspections using the trace and posterior density plots. A sample of such plots obtained for the parameters of 2-p Weibull and 3-p Weibull distributions from the Bayesian simulations are presented in Figures 1-4 for sites 3 and 4. The plots for the remaining sites are presented in Figures A1-A10 in Appendix 2.

In Figure 1, trace plots (left) of 2-p Weibull for site 3 show 'fat hairy caterpillar' appearance which is indicative of a random scatter around the stable mean of the shape parameter $k = 1.947640$ within 95% credible region (1.936812, 1.958478) and the scale parameter $A = 7.017671$ within 95% credible region (6.992180, 7.042895). The density plots (right) also display smooth curves of these simulated values for both the parameters. Thus, the plots clearly indicate the convergence of the simulations of the Bayesian estimates presented in Table 5.

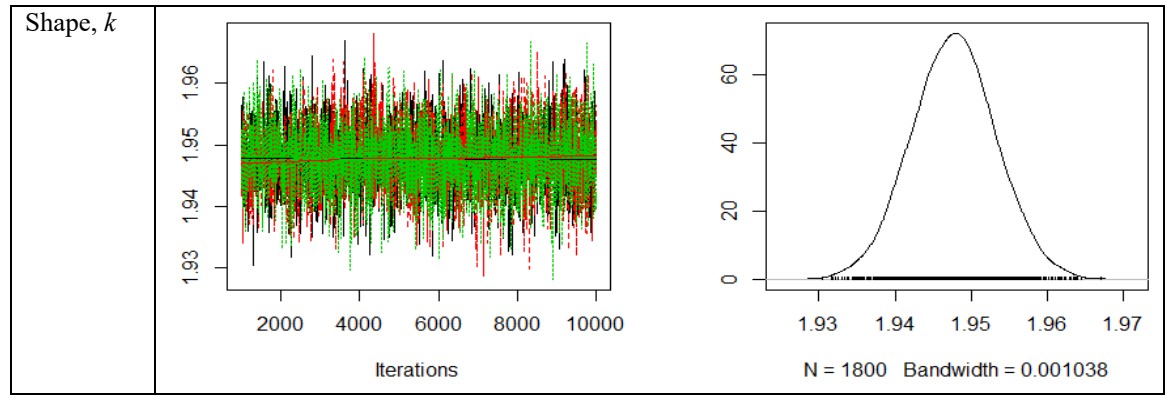



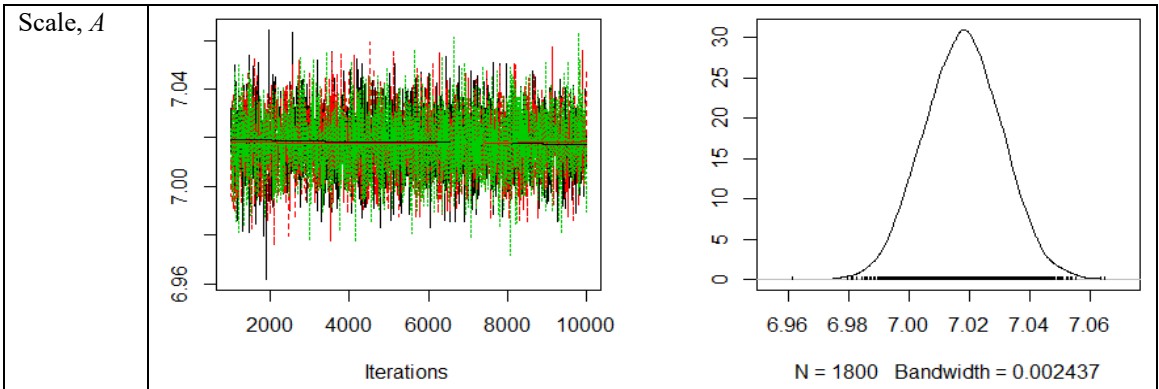

**Figure 1: Trace and posterior density plots for site 3 (2-p Weibull).**

In Figure 2, the trace plots (left) of 3-p Weibull also show 'fat hairy caterpillar' appearance which is indicating a random scatter around the stable mean of the shape parameter $k = 2.635850$ within 95% credible region (2.604492, 2.667974) and the scale parameter $A = 8.804653$ within 95% credible region (8.718409, 8.892835). The shift parameter shows a random scatter around the stable mean of $\theta = -1.546999$ within 95% credible region (-1.624995, -1.472661), which is far away from zero and it reveals the appropriateness of using 3-p Weibull. The density plots (right) also display smooth curves of these simulated values for all the parameters. Thus, the plots clearly indicate the convergence of the simulations of Bayesian estimates presented in Table 5.

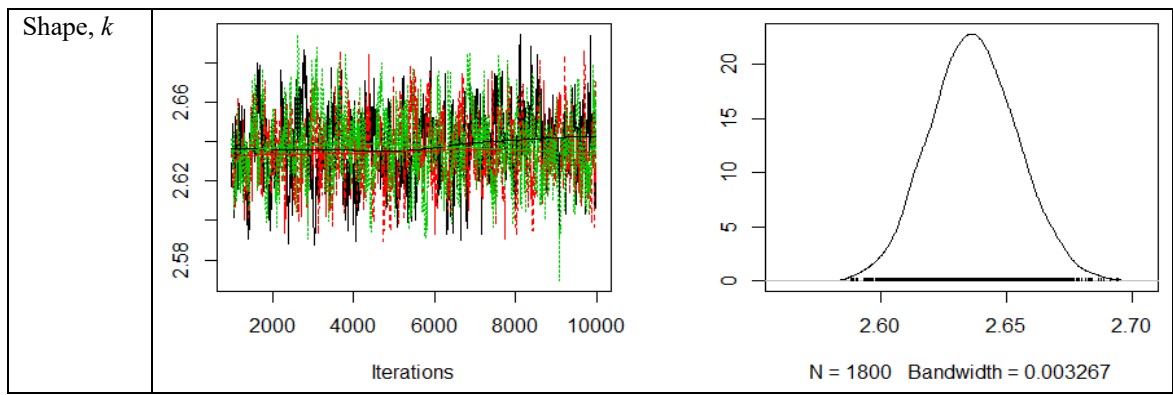





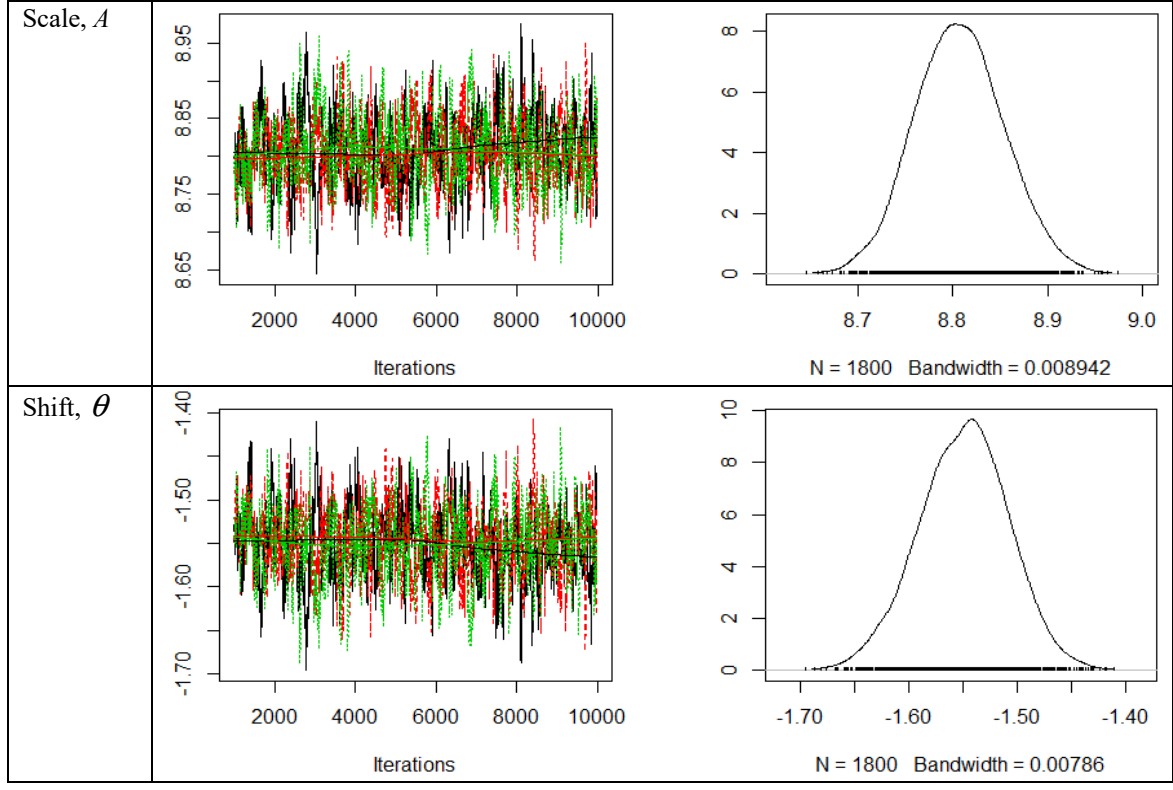

**Figure 2: Trace and posterior density plots for site 3 (3-p Weibull).**

Similarly, from the Figures 3-4, the trace plots (left) for site 4 indicate random scatter around the stable mean of the shape parameter $k$ and the scale parameter $A$, which are within their 95% credible regions for both 2-p Weibull and 3-p Weibull distributions. The shift parameter in Figure 4 also shows a random scatter around the stable mean of $\theta$ = -0.055714 within 95% credible region (-0.169674, 0.031573), which includes zero and it reveals the appropriateness of using 2-p Weibull. The density plots (right) also display smooth curves of these simulated values for all the parameters. Thus, the plots clearly indicate

the convergence of the simulations of Bayesian estimates presented in Table 5.

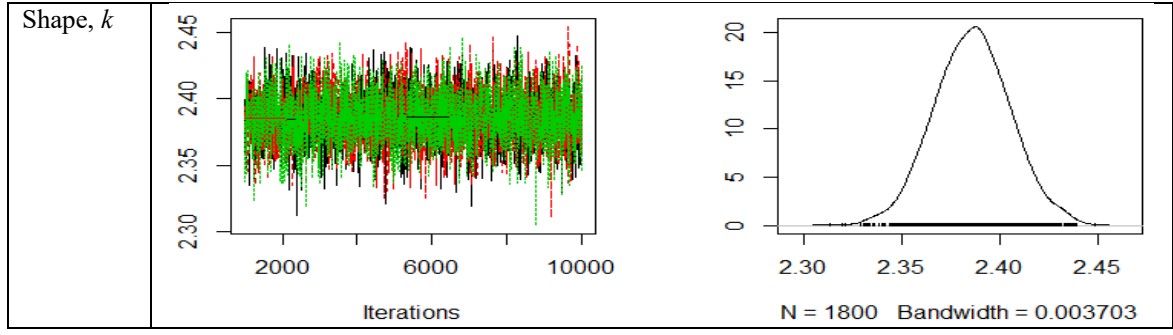





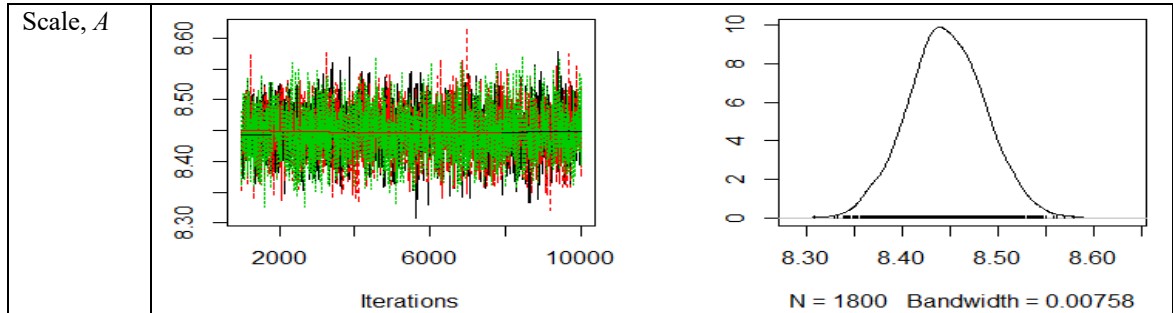

**Figure 3: Trace and posterior density plots for site 4 (2-p Weibull).**

**Figure 4: Trace and posterior density plots for site 4 (3-p Weibull).**




## 8 Discussion

In Section 7, the results for the goodness of fit for the wind speed distributions at seven different sites were presented. Results showed that the 3-p Weibull distribution is a better fit for wind speeds at all the sites investigated, except the sites 4 and 6, in the Equatorial region. The 2-p Weibull distribution may be a better fit for wind speeds data at the sites 4 and 6 as the shift

parameter $\theta$ in the 3-p Weibull was found to be zero as detected in Bayesian estimate. As discussed earlier, the 2-p Weibull distribution can be considered a special case of 3-p Weibull distribution.

In this section, results of further investigations carried out to explain the difference between the performance of the two distributions are presented and discussed. Referring to the percentage of lowest wind speed (0-1 m/s) presented in Table 6, the results clearly show that the wind distributions of the sites (sites 1, 2, 3, 5 and 7) that have high percentage (0.79% - 7.04%)

of lower (or closer to null) wind speeds perfectly fit the 3-p Weibull distribution. The shift parameter in the simulations is also found to be significant i.e. $\theta \neq 0$ for these sites. On the other hand, the 2-p Weibull distribution was a better fit for wind speed distributions at Sites 4 and 6, where the percentage of low wind speed was smaller (< 0.78%). Similar findings were reported by Wais (2017).

Moreover, histograms presented in Figures 5-11 for wind distributions at sites 1 to 7 show different shapes, indicating a

variation in skewness. Thus, another reason for fitting a better distribution is the skewness of the wind speed distribution. The skewness ($\gamma_1$) is a measure of the asymmetry of the wind speed distribution about its mean, which is defined for a sample of $n$ values as:

$$\gamma_1 = \frac{m_3}{s^3} \tag{30}$$

where, s = standard deviation and $m_3 = \frac{1}{n}\sum_{i=1}^{n}(U-\overline{U})^3$ is the third central moment. A normal distribution is symmetrical

and has $\gamma_1 = 0$. If $\gamma_1$ is negative, the distribution is left skewed whereas a positive $\gamma_1$ indicates a right skewed distribution. Since the Weibull distribution is a right skewed one, $\gamma_1$ is expected to be positive.

Table 6 presents the mean ($\overline{U}$), standard deviation ($s$) and skewness ($\gamma_1$) of the wind speed data at each site. It shows that the wind speed distributions of sites 4 and 6 have higher skewness compared to sites 1, 2, 3, 5 and 7.

**Table 6: Percentage lowest wind speed, mean, SD and best Weibull distribution for the seven sites.**

| Site | % lowest speed (0-1 m/s) | Site mean wind speed ($\overline{U}$) | SD ($s$) | Skewness ($\gamma_1$) | Fitted distribution |
|------|--------------------------|---------------------------------------|----------|-----------------------|---------------------|
| 1 | 1.71 | 5.35 | 2.22 | 0.28 | 3-p Weibull |



| 2 | 1.87 | 5.83 | 2.29 | 0.09 | 3-p Weibull |
| 3 | 7.04 | 6.29 | 3.20 | 0.20 | 3-p Weibull |
| 4 | 0.71 | 7.50 | 3.31 | 0.62 | 2-p Weibull |
| 5 | 0.79 | 7.31 | 3.20 | 0.22 | 3-p Weibull |
| 6 | 0.78 | 6.97 | 2.95 | 0.51 | 2-p Weibull |
| 7 | 3.08 | 6.32 | 3.04 | 0.29 | 3-p Weibull |

Thus, the results reveal that the 3-p Weibull distribution is a better fit for wind speed data with both: greater frequency of low wind speeds (0-1 m/s) and low skewness, compared to a 2-p Weibull distribution. The Bayesian analysis also confirms that the wind speed data with a smaller percentage of low speeds fitted better as a 2-p Weibull distribution than the 3-p Weibull distribution as its shift parameter $\theta$ was found to be zero.

To reiterate, this research is aimed at comparing the goodness of fit of both 2-p and 3-p Weibull distributions and to compare the performance of frequentist MLE and Bayesian methods for the estimation of Weibull parameters. Therefore, a comparison study of the four methods is conducted by:

1.  Fitting of 2-p Weibull distribution with an MLE estimate – MLE.2p

2.  Fitting of 3-p Weibull distribution with an MLE estimate – MLE.3p

3.  Fitting of 2-p Weibull distribution with a Bayesian estimate – BAYESIAN.2p

4.  Fitting of 3-p Weibull distribution with a Bayesian estimate – BAYESIAN.3p

The wind speed bins and the Weibull distribution curves obtained using these four methods are illustrated in Figures 5-11. The figures show that for all the sites, except the sites 4 and 6, the 3-p Weibull distributions are closer to the histograms than the 2-p Weibull distributions. The Weibull distribution curves of Bayesian estimates are also closer to the histograms than MLE estimates. Thus, the 3-p Weibull distribution is a better fit for the wind speed data compared to the 2-p Weibull distribution (Aukitino et al., 2017).



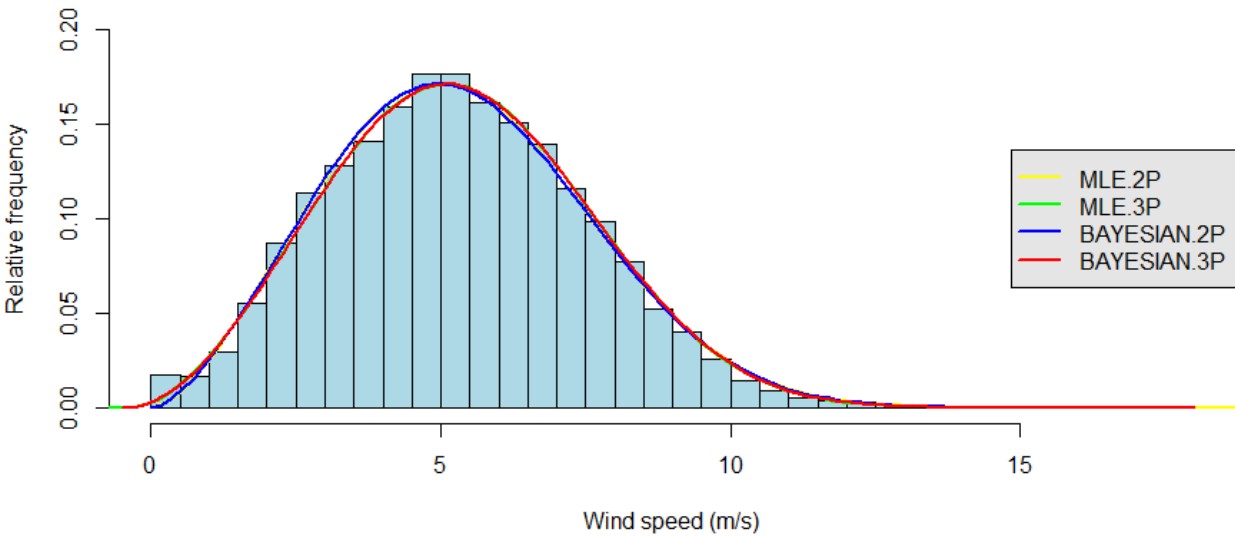

**Figure 5: 2-p and 3-p Weibull curves by four methods and histogram of the observed wind speeds at site 1.**

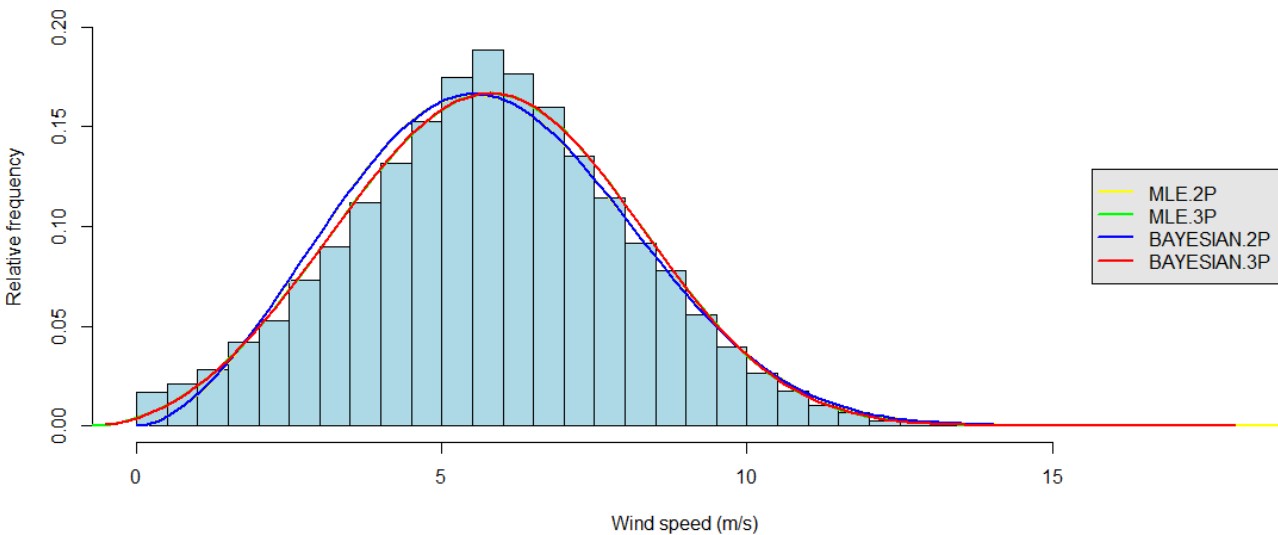

**Figure 6: 2-p and 3-p Weibull curves by four methods and histogram of the observed wind speeds at site 2.**



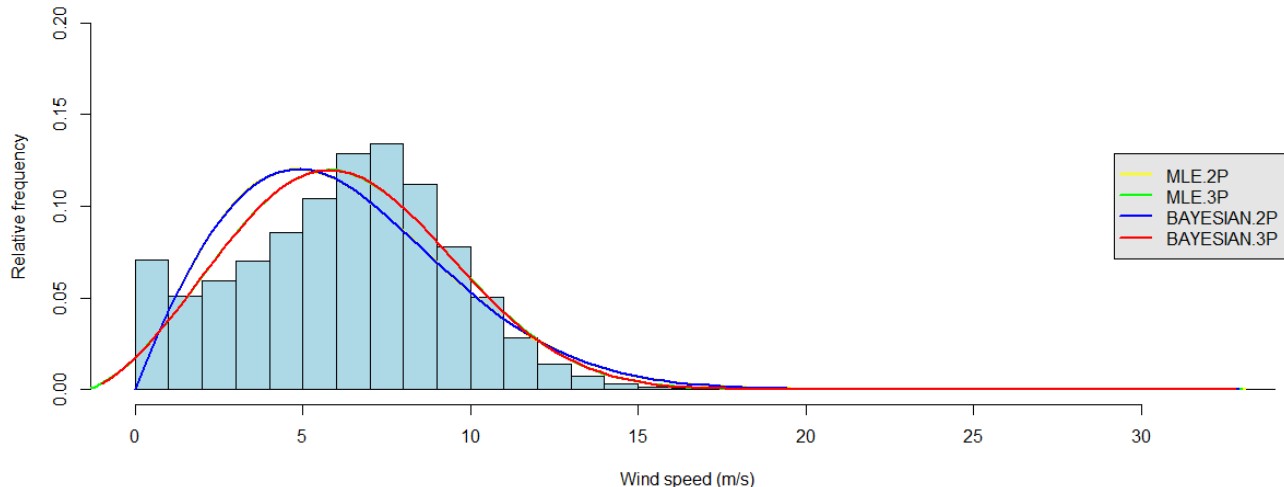


**Figure 7: 2-p and 3-p Weibull curves by four methods and histogram of the observed wind speeds at site 3.**

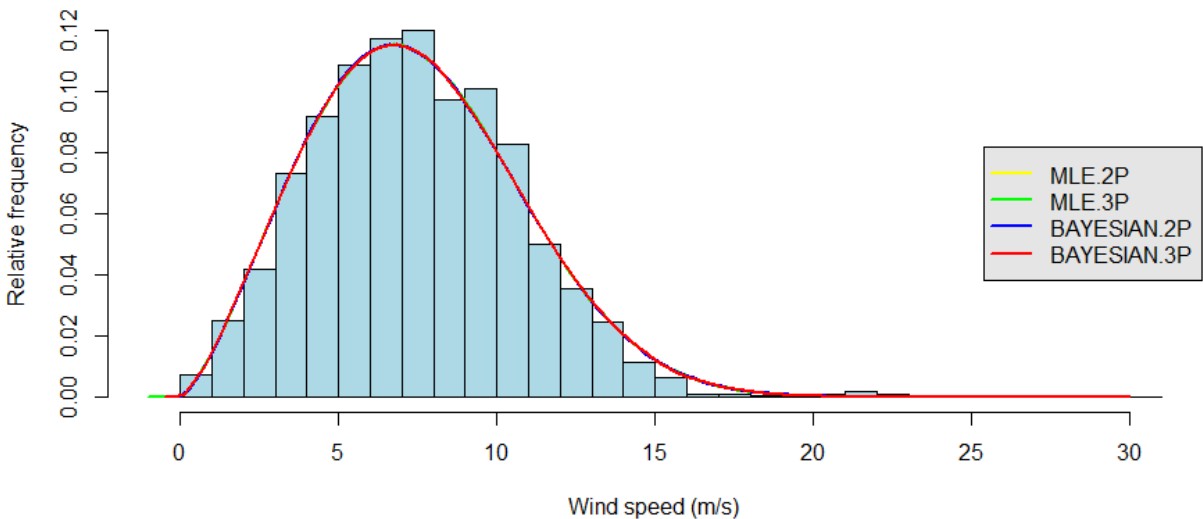

**Figure 8: 2-p and 3-p Weibull curves by four methods and histogram of the observed wind speeds at site 4.**


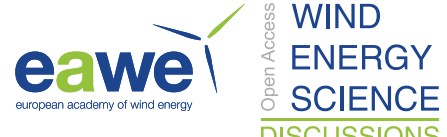

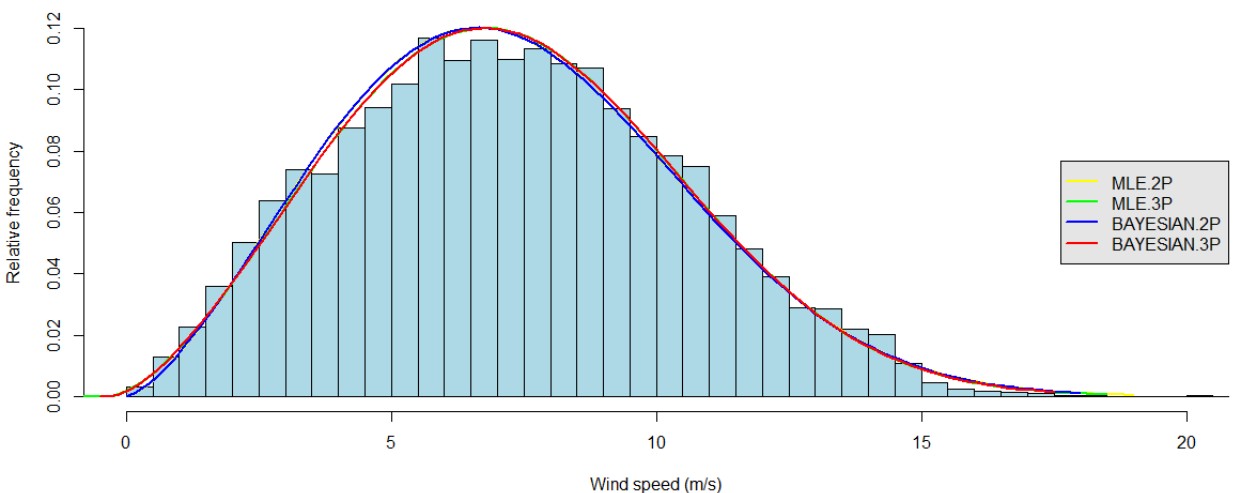

**Figure 9: 2-p and 3-p Weibull curves by four methods and histogram of the observed wind speeds at site 5.**


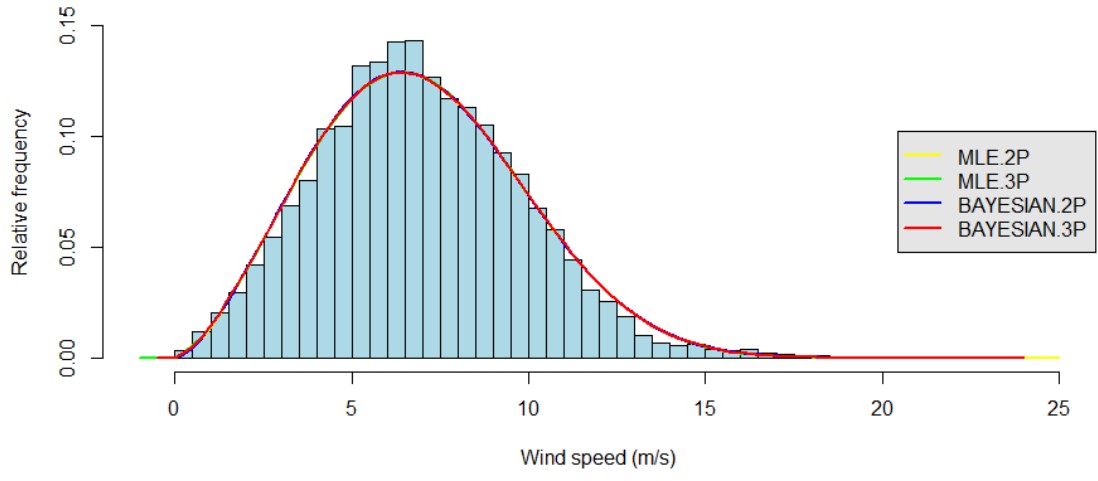

**Figure 10: 2-p and 3-p Weibull curves by four methods and histogram of the observed wind speeds at site 6.**





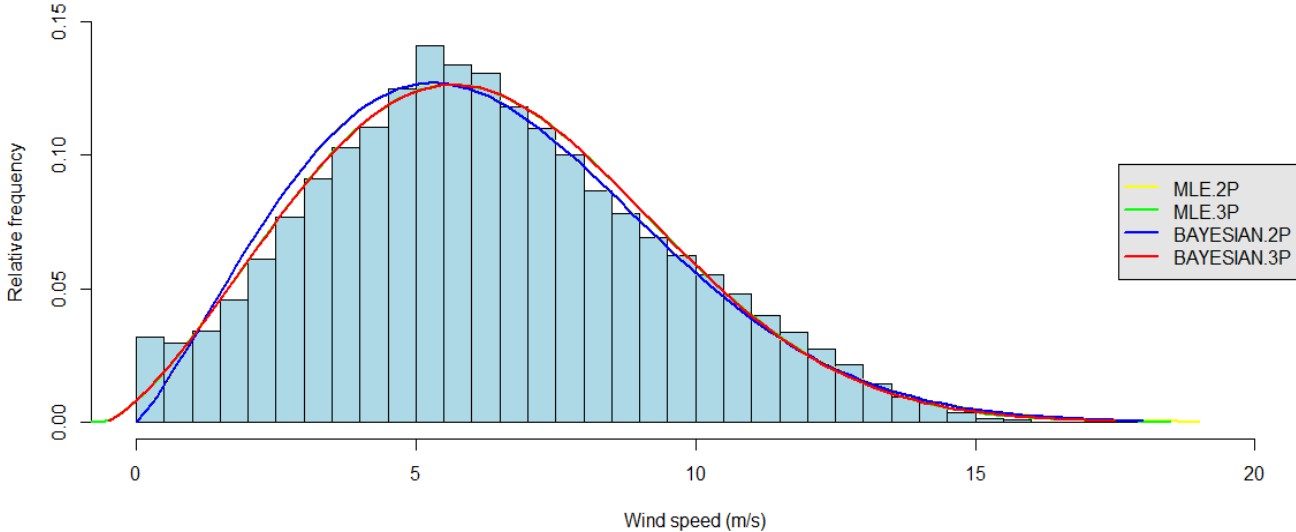


**Figure 11: 2-p and 3-p Weibull curves by four methods and histogram of the observed wind speeds at site 7.**

Figures 5, 6 and 9 show symmetric Weibull distribution curves for the sites 1, 2 and 5. The mean wind speed is close to the centre and the percentage of the lowest wind speed (0 – 1 m/s) is relatively small to moderate at these sites, as can be seen in Table 6. The distribution curves clearly indicate that the BAYESIAN.3P Weibull distribution fits the data best, which is also

supported by the Goodness-of-fit measures presented in Table 7.

However, figures 7 and 11 show non-symmetric distributions of wind speeds for the sites 3 and 7. The distribution curves clearly indicate that the BAYESIAN.3P Weibull distribution fits better to the data in these sites. It can be noted that these sites recorded very high percentage of low wind speeds (0-1 m/s) as seen in Table 6.

On the other hand, figures 8 and 10 show relatively less symmetric wind speed distribution for site 4 and 6 compared to the

other sites with clearly high skewness as seen in Table 6. For these sites, the percentage of the lowest wind speeds is smaller, hence the Bayesian 2-p distribution fits the wind speed data well. Although the four Weibull distribution curves seem to be coinciding, the BAYESIAN.2P distribution curves lie above the 3-p one before the peak and below the 3-p curve after the peak. From Table 4, it can be seen that the value of the shift parameters $\theta$ are the smallest (close to zero) for these sites, hence the 3-p Weibull distribution becomes 2-p distribution.

To evaluate the performance of the four methods, the five statistical goodness of fit measures discussed in Section 6: $R^2$, RMSE, COE, MAE and MAPE are estimated and compared. The results of the estimation of these measures for the wind data from all the the seven sites are presented in Table 7, which clearly reveal that BAYESIAN.3P is the most efficient method for the sites 2, 3, 5 and 7 as it gives highest $R^2$ value and the least RMSE, MAE and MAPE values. This indicates that the 3-p Weibull with Bayesian estimates is a better method for wind energy assessment at these sites. Moreover, the method equally

performs with MLE.3p for the site 1. Whereas, BAYESIAN.2P performs better for the sites 4 and 6 producing highest $R^2$ value





and the least COE and RMSE values, which indicates 2-p Weibull with Bayesian estimates is a better method for assessment at these two sites which have the lowest occurrence of smallest wind speeds in the range of 0-1 m/s.

**Table 7: Goodness-of-fit measures for the four methods.**

| Site | Method | $R^2$ | COE | RMSE | MAE | MAPE |
|------|--------|-------|-----|------|-----|------|
| 1 | MLE.2P | 0.9982 | **0.9936** | 0.0983 | 0.0421 | 1.3052 |
| | MLE.3P | **0.9983** | 1.0011 | **0.0975** | **0.0239** | **0.8269** |
| | BAYESIAN.2P | 0.9982 | **0.9936** | 0.0983 | 0.0421 | 1.3053 |
| | BAYESIAN.3P | **0.9983** | 1.0011 | **0.0975** | **0.0239** | **0.8269** |
| 2 | MLE.2P | 0.9949 | 1.0048 | 0.1655 | 0.1114 | 3.4259 |
| | MLE.3P | **0.9971** | **1.0034** | 0.1234 | 0.0661 | 1.7917 |
| | BAYESIAN.2P | 0.9949 | 1.0048 | 0.1655 | 0.1114 | 3.4259 |
| | BAYESIAN.3P | **0.9971** | 1.0041 | **0.1227** | **0.0645** | **1.7748** |
| 3 | MLE.2P | 0.9696 | 0.9823 | 0.5675 | 0.4596 | 10.7507 |
| | MLE.3P | **0.9755** | 1.0681 | 0.5219 | 0.3029 | 7.9253 |
| | BAYESIAN.2P | 0.9696 | **0.9822** | 0.5677 | 0.4597 | 10.7516 |
| | BAYESIAN.3P | **0.9755** | 1.0679 | **0.5216** | **0.3027** | **7.9149** |
| 4 | MLE.2P | **0.9787** | 1.0512 | 0.4983 | 0.1546 | 1.9778 |
| | MLE.3P | 0.9783 | 1.0523 | 0.5025 | 0.1522 | 1.9013 |
| | BAYESIAN.2P | **0.9787** | **1.0511** | **0.4981** | 0.1544 | 1.9724 |
| | BAYESIAN.3P | 0.9783 | 1.0524 | 0.5036 | **0.1518** | **1.8860** |
| 5 | MLE.2P | 0.9983 | **1.0180** | 0.1371 | 0.1031 | 2.0079 |
| | MLE.3P | 0.9987 | 1.0186 | 0.1109 | 0.0787 | 1.5343 |
| | BAYESIAN.2P | 0.9983 | 1.0181 | 0.1372 | 0.1033 | 2.0074 |
| | BAYESIAN.3P | **0.9989** | 1.0183 | **0.1108** | **0.0785** | **1.5295** |
| 6 | MLE.2P | **0.9911** | 1.0173 | 0.2809 | **0.1251** | 1.9635 |
| | MLE.3P | 0.9909 | 1.0181 | 0.2852 | 0.1257 | 1.9187 |
| | BAYESIAN.2P | **0.9911** | **1.0169** | **0.2806** | 0.1253 | 1.9673 |
| | BAYESIAN.3P | 0.9909 | 1.0180 | 0.2854 | 0.1258 | **1.9162** |
| 7 | MLE.2P | 0.9973 | **0.9991** | 0.1607 | 0.1126 | 3.4174 |
| | MLE.3P | 0.9987 | 1.0152 | 0.1069 | 0.0718 | 2.0482 |
| | BAYESIAN.2P | 0.9973 | 0.9992 | 0.1609 | 0.1130 | 3.4244 |
| | BAYESIAN.3P | **0.9988** | 1.0152 | **0.1068** | **0.0717** | **2.0476** |


**Assessment of wind power and energy:**

To estimate the turbine productivity, it is necessary to calculate the wind power and the AEP from each fitted model. If the density of the air $\rho = 1.16$ kg/m³ (based on the average temperature in the region) and the turbine rotor diameter $D = 32$ m, the expected annual wind energy produced from each site is determined using equations (22-25) for both 2-p and 3-p Weibull distributions. Table 8 shows estimated results for wind power and AEP.

The relative error of the estimated power shown in Table 18 is calculated as follows:





$$\text{Re}\,lative\ error\ (\%),\ RE\ =\ \frac{Estimated\ power\ -\ Actual\ power}{Actual\ power}\times100\%$$

If the RE is close to zero, the method estimates the parameter accurately. However, a positive RE implies over-estimation and a negative RE implies under-estimation by the method. From the values of Power, RE and AEP presented in Table 8, we observed that the BAYESIAN.2P and BAYESIAN.3P are found to be the most efficient methods for estimating power and AEP. The BAYESIAN.3P predicts the most accurate wind power and AEP for the sites 1, 2, 3, 5 and 7, which is very close to the actual power of these sites with smaller RE compared to the other methods. On the other hand, the BAYESIAN.2P provides the best estimate of power and AEP for the sites 4 and 6 with smaller relative error.

**Table 8: Estimated power, relative error in power estimate and annual energy production.**

| Site | Methods | Power (kW) | RE (%) | AEP (kWh) |
|------|---------|-----------|--------|-----------|
| 1 | Actual | 108457 | 0.00 | 950083 |
|   | MLE.2P | 110306 | 1.70 | 966279 |
|   | MLE.3P | 110061 | 1.47 | 964134 |
|   | BAYESIAN.2P | 110298 | 1.70 | 966206 |
|   | **BAYESIAN.3P** | **110051** | **1.47** | **964047** |
| 2 | Actual | 133525 | 0.00 | 1169683 |
|   | MLE.2P | 136039 | 1.88 | 1191705 |
|   | MLE.3P | 135765 | 1.68 | 1189301 |
|   | BAYESIAN.2P | 136030 | 1.88 | 1191623 |
|   | **BAYESIAN.3P** | **135753** | **1.67** | **1189195** |
| 3 | Actual | 206184 | 0.00 | 1806168 |
|   | MLE.2P | 220464 | 6.93 | 1931263 |
|   | MLE.3P | 209474 | 1.60 | 1834996 |
|   | BAYESIAN.2P | 220556 | 6.97 | 1932074 |
|   | **BAYESIAN.3P** | **209458** | **1.59** | **1834856** |
| 4 | Actual | 322492 | 0.00 | 2825032 |
|   | MLE.2P | 319732 | -0.86 | 2800855 |
|   | MLE.3P | 319627 | -0.89 | 2799928 |
|   | **BAYESIAN.2P** | **319812** | **-0.83** | **2801557** |
|   | BAYESIAN.3P | 319591 | -0.90 | 2799619 |
| 5 | Actual | 290717 | 0.00 | 2546683 |
|   | MLE.2P | 291809 | 0.38 | 2556248 |
|   | MLE.3P | 290928 | 0.07 | 2548528 |
|   | BAYESIAN.2P | 291696 | 0.34 | 2555257 |
|   | **BAYESIAN.3P** | **290889** | **0.07** | **2548188** |
| 6 | Actual | 249192 | 0.00 | 2182919 |
|   | MLE.2P | 247792 | -0.56 | 2170656 |
|   | MLE.3P | 247770 | -0.57 | 2170468 |
|   | **BAYESIAN.2P** | **247848** | **-0.54** | **2171151** |
|   | BAYESIAN.3P | 247808 | -0.56 | 2170799 |
| 7 | Actual | 204938 | 0.00 | 1795257 |
|   | MLE.2P | 207543 | 1.27 | 1818077 |
|   | MLE.3P | 204781 | -0.08 | 1793882 |
|   | BAYESIAN.2P | 207487 | 1.24 | 1817586 |
|   | **BAYESIAN.3P** | **204790** | **-0.07** | **1793960** |




Thus, the comparison of results based on the goodness of fit and the power estimation at different sites in the Equatorial region show that the **3-p Weibull distribution with Bayesian estimates is the method to be used for wind energy resource assessments**. If any sites have lower occurrences of the lower wind speeds, then the shift parameter $\theta$ in the 3-p Weibull distribution will be close to or equal to zero, which will be the 2-p Weibull distribution with Bayesian estimates. Another

advantage of Bayesian approach is that it will reduce the need for long-term measurements for assessing the wind power potential of a site. This is possible with the integration of prior information with short-term data from a candidate site or historical data from one neighboring survey station.

## 9 Conclusion

Knowledge of correct statistical distribution of wind speeds at a given site is very important for accurate wind resource

assessment. Some sites provide high uncertainty while fitting the traditional two-parameter Weibull distributions to wind speed data and warrant the need to explore distributions that characterize wind speeds better, such as the three-parameter Weibull distribution which is also a generalized form of two-parameter Weibull distribution with an additional non-zero shift parameter. In this study, investigation of wind characteristics and wind energy potential are carried out at different locations in the Equatorial region ranging from 1° N to 21° South of the Equator, of which three sites are in Fiji and one each from Cooks

Islands, Tonga, Kiribati and Vanuatu, respectively. The wind speed data at these seven sites were tested for the best model between the two-parameter and three-parameter Weibull distributions. Furthermore, as there is no unique method that characterizes wind data perfectly, it is also imperative to have the knowledge of the best method of estimation for the parameters of wind speed distribution at a given site. In this work, we also introduced a novel approach by using the Bayesian method for estimating parameters of wind speed distributions at the seven sites selected for testing the method. Then, a

comparison study was conducted for the robust performance of the proposed Bayesian method with the popular frequentist MLE method. Finally, the results suggest that the three-parameter Weibull distributions should be used for estimating the wind energy potential irrespective of the location. When the wind distribution has frequent low wind speeds and is less skewed, a three-parameter Weibull distribution is found to be a better fit. On the other hand, when the wind distribution has less frequent low speeds and highly skewed, the two-parameter Weibull distribution which is a special case of three-parameter distribution

with zero shift parameter is found to be more appropriate. The results also indicate that the Bayesian approach provides more accurate results while characterizing wind speeds and can be proposed as a better alternative for estimating Weibull parameters. The proposed method can be incorporated in the popular software packages such as WAsP used for wind resource assessment and for planning wind energy projects.

## Code availability

Model code is provided in Appendix 1. Full code can be made available for a very valid request.

## Data availability



Sample data files are provided as Appendix 3 (a-c). Full data will be made available upon request and after approval from the respective Governments.

## Author contributions

MGMK: Conceptualization of new method, writing the code, simulations and draft paper; MRA: Project approval, data acquisition, compilation, validation and review of manuscript.

## Declaration of competing interests

The authors declare that they have no conflict of interest with respect to the research, authorship, and/or publication of this article.

## 540 Funding

Funds for carrying out part of this work were provided by Korea International Cooperation Agency (KOICA) under its East-Asia Climate Partnership program. The project number was 2009-00042.

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



# Appendix 1


## Model Code:

`JAGS` model specifications to perform the Bayesian fit using `R2jags` package are provided below.


### 1. Model specification for 2-parameter Weibull distribution

```
cat("
    model{
    # Likelihood
    for (i in 1:length(U)){
    p[i] <- dweib(U[i],shape, lambda);
    observed[i] ~ dbern(p[i]);
    }

    # Priors
    shape ~ dunif(0,4)  # After a series of trial and error guesses
    scale ~ dunif(0,100)
    lambda <- pow(1/(scale), shape)
    }
    ", file="weibull_model_2p.txt")
```




### Calling JAGS from R

```
jags.fit <- jags(data, inits, parameters.to.save, n.iter=10000,
model.file="weibull_model_2p.txt",n.chains = 3, n.burnin = 1000, n.thin=5)
```


### 2. Model specifications for the 3-parameter Weibull distribution

```
cat("
    model{
    # Likelihood
    for (i in 1:length(U)){
    p[i] <- dweib(U[i] - shift ,shape, lambda) / 1000
    observed[i] ~ dbern(p[i]);
    }

    # Priors
    shape ~ dunif(0,4)
    scale ~ dunif(0,31)
    shift ~ dunif(-1, 1)
    lambda <- pow(1/(scale), shape)
    }
    ", file="weibull_model_3p.txt")
```









# Appendix 2

**Trace and posterior density plots that show the convergence of Bayesian models for sites 1, 2, 5, 6 and 7.**


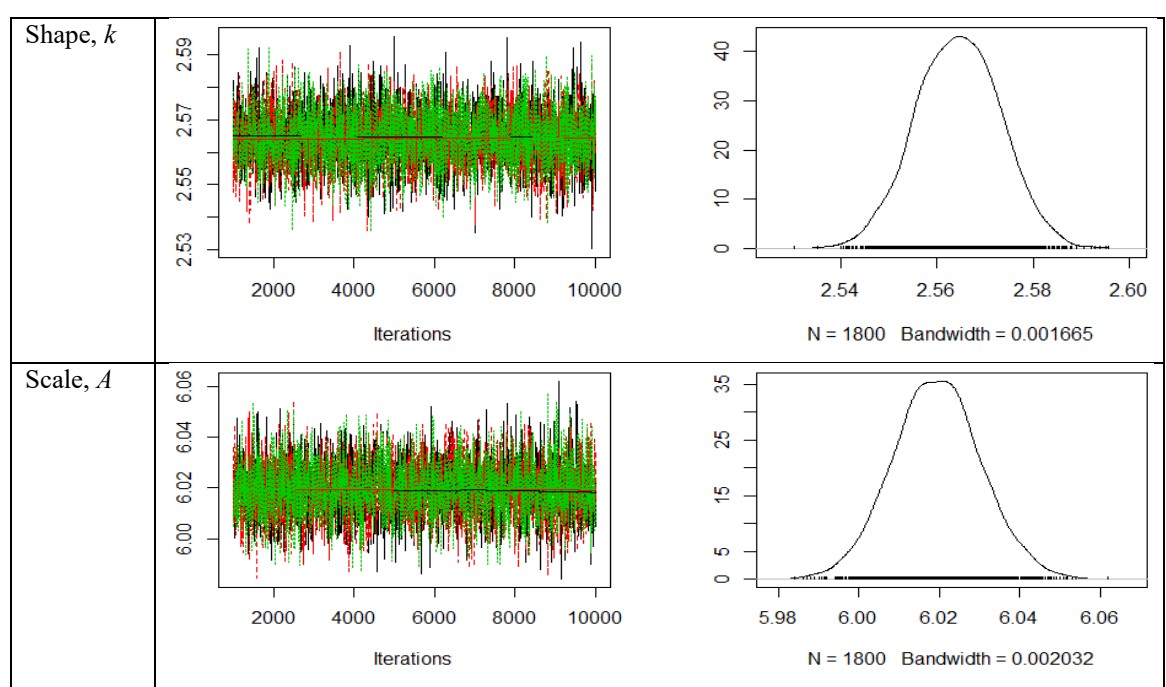

**Figure A1: Trace and posterior density plots for site 1 (2-p Weibull).**

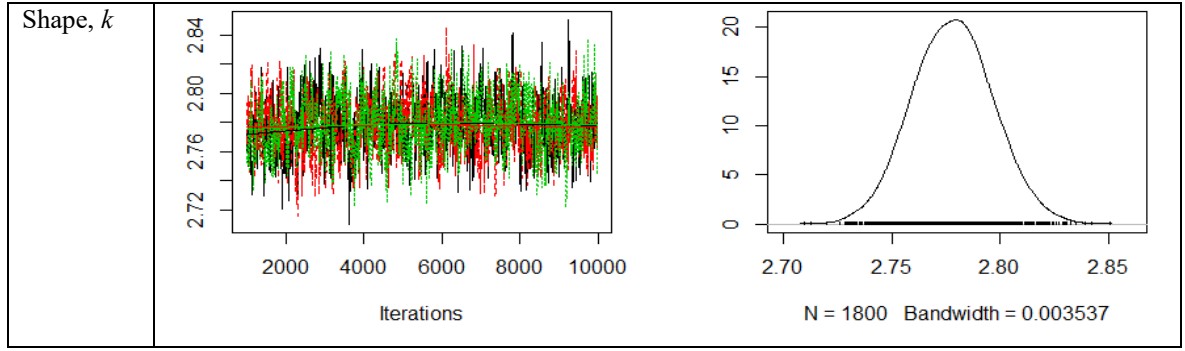



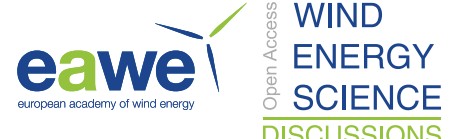

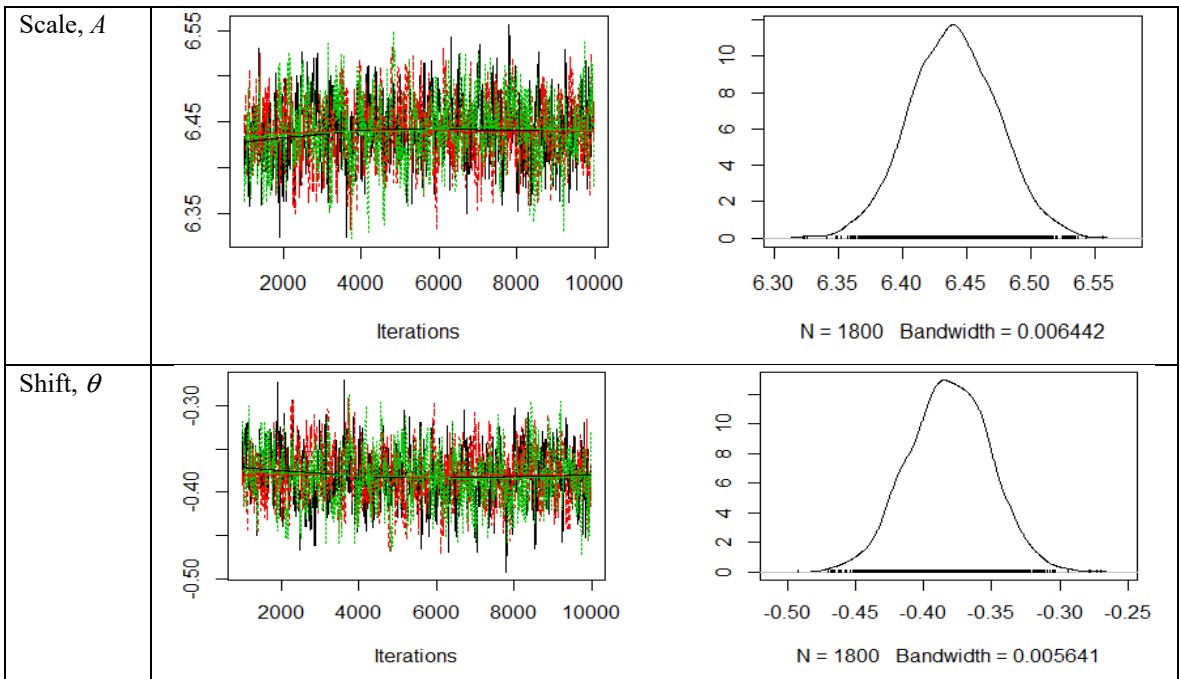

**Figure A2: Trace and posterior density plots for site 1 (3-p Weibull).**


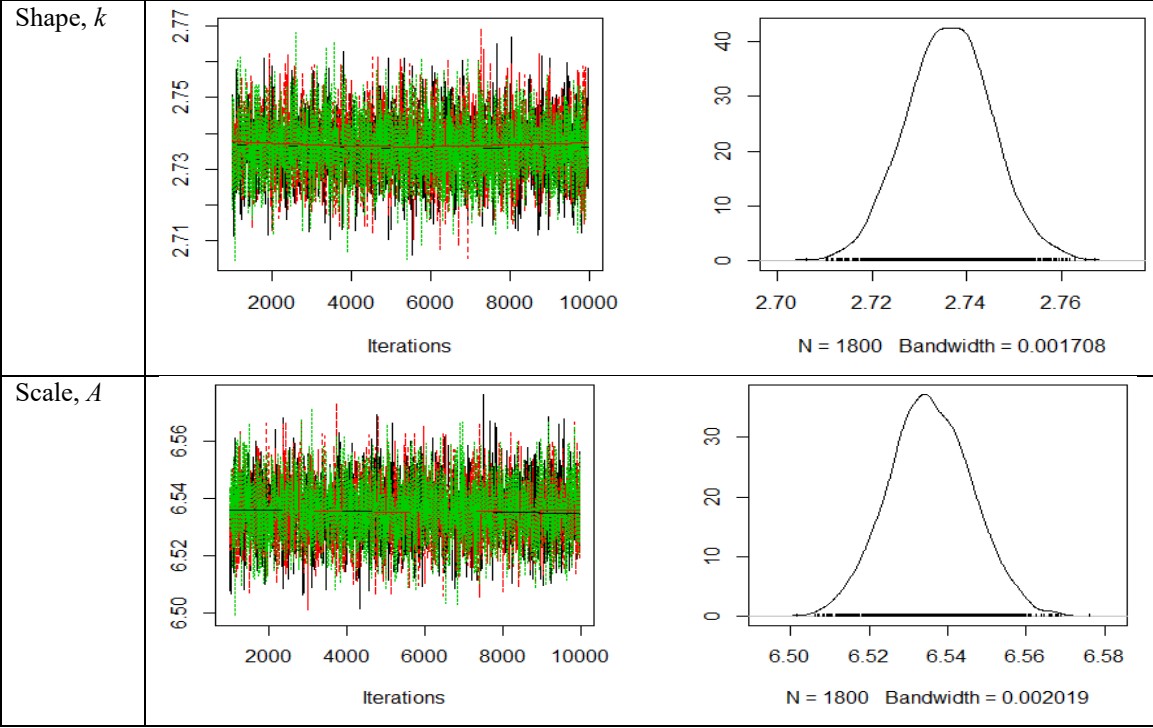

**Figure A3: Trace and posterior density plots for site 2 (2-p Weibull).**







**Figure A4: Trace and posterior density plots for site 2 (3-p Weibull).**





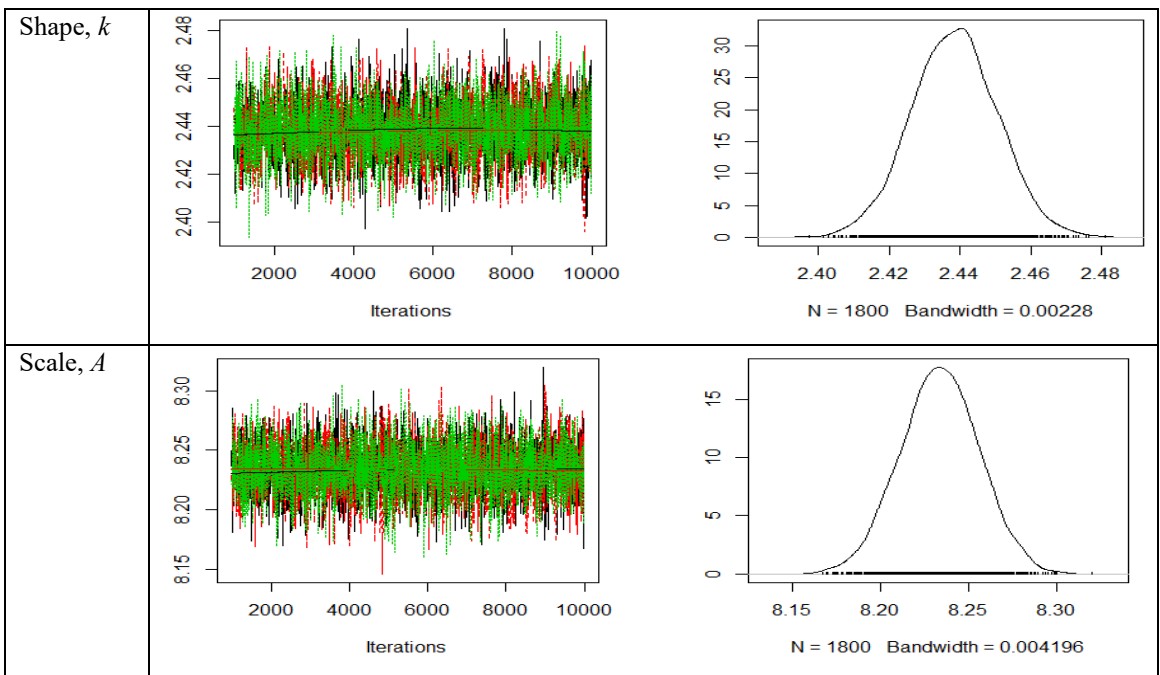

**Figure A5: Trace and posterior density plots for site 5 (2-p Weibull).**


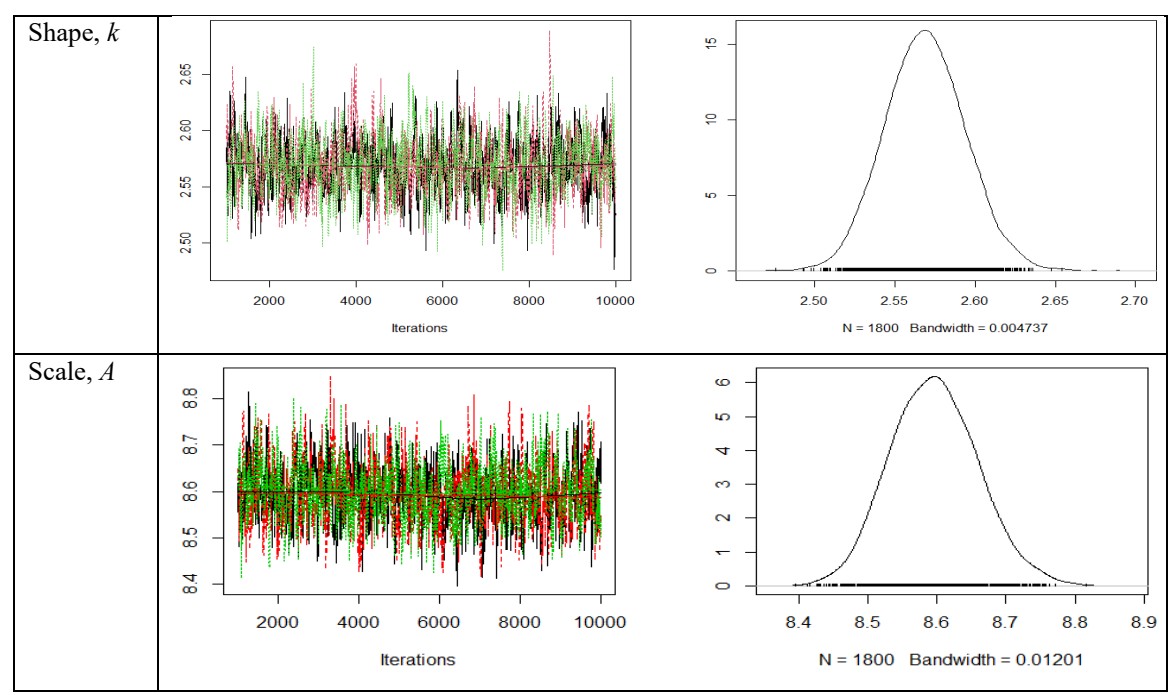





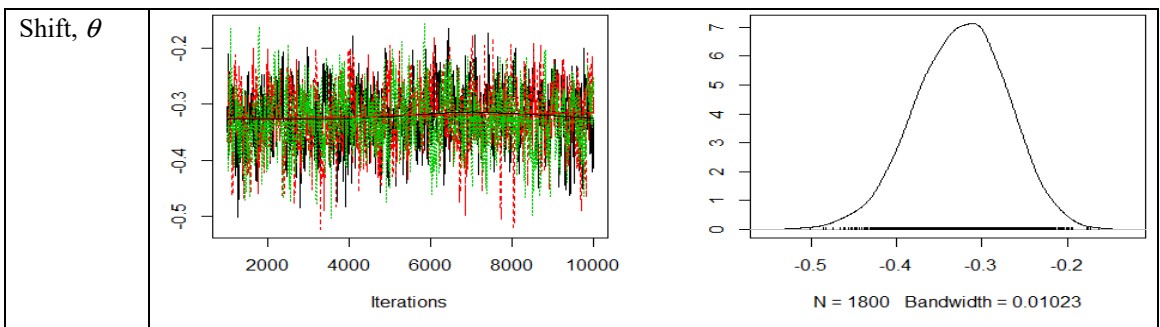

**Figure A6: Trace and posterior density plots for site 5 (3-p Weibull).**

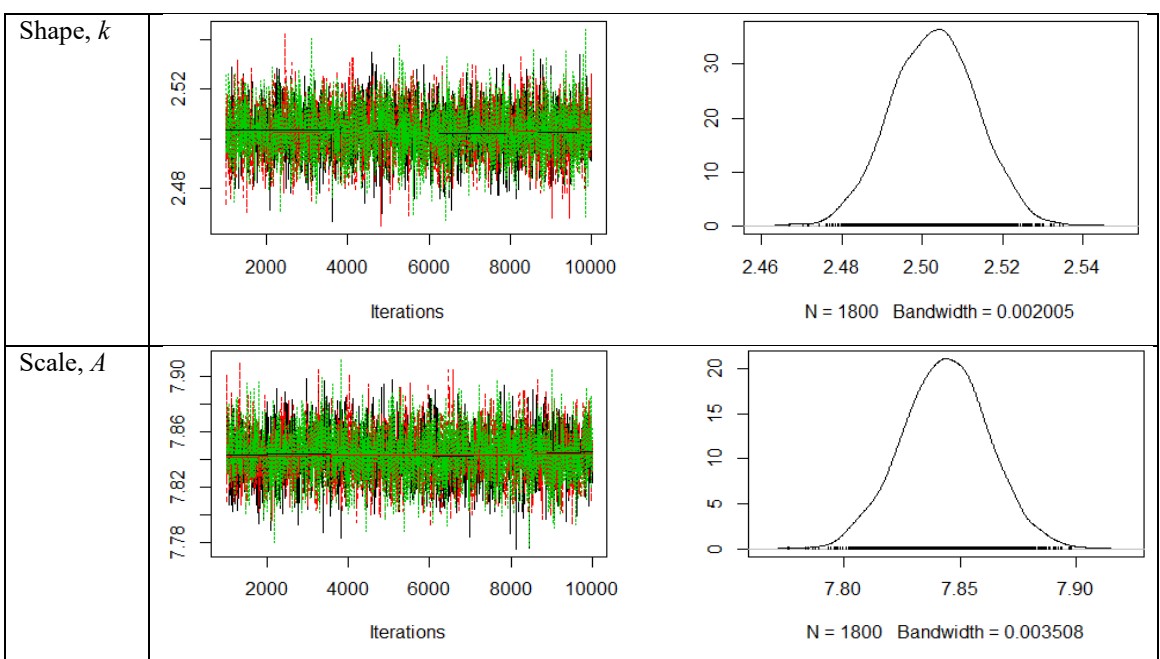

**Figure A7: Trace and posterior density plots for site 6 (2-p Weibull).**


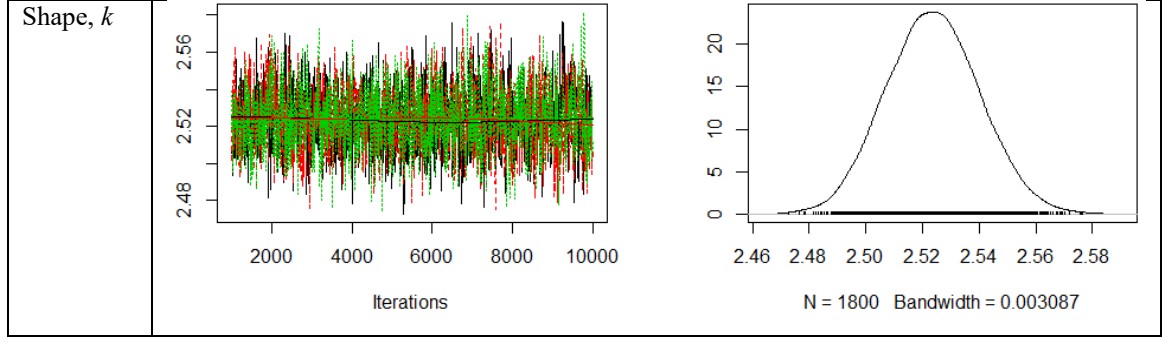





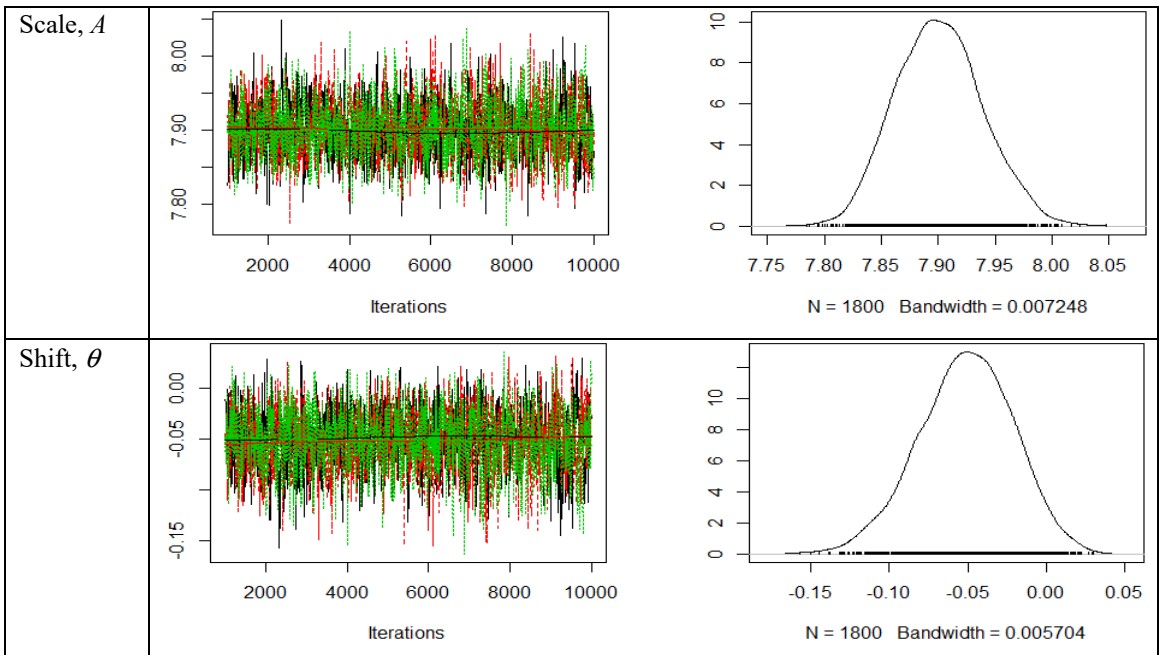

**Figure A8: Trace and posterior density plots for site 6 (3-p Weibull).**

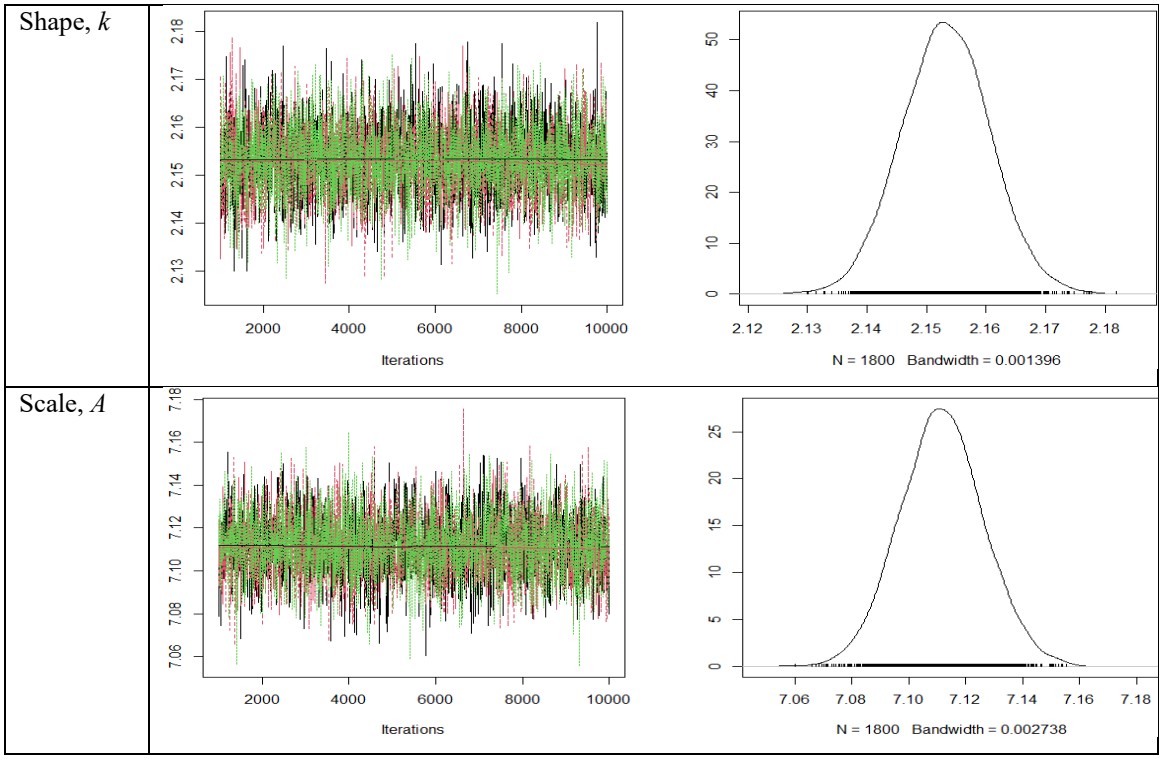

**Figure A9: Trace and posterior density plots for site 7 (2-p Weibull).**





**Figure A10: Trace and posterior density plots for site 7 (3-p Weibull).**