# Peer review of "Bayesian method for estimating Weibull parameters for wind resource assessment in the tropical region: a comparison between twoparameter and three-parameter Weibull distributions"

_Wind Energy Science, 2022_

## Author Comment (AC1)

**Response to Reviewers' Comments (WES-2022-70)**

**Reviewer 1:**

**GENERAL COMMENTS:**
This paper examines a genuine issue in the science of wind energy and is thus in scope for this journal (Wind Energy Science).  There are certainly places with significant periods of calm (wind speed < 1m/s) but in which wind power may nevertheless be an option worth considering.  I know of  some such sites in Fiji and elsewhere in the Pacific Islands. There is something of a cottage industry in fitting [2-parameter] Weibull distributions to measured wind speed at particular sites around the world, with at least 400 papers published since 2006 with such a subject, according to the Scopus database. But, as the authors point out, a  standard 2-parameter Weibull distribution of wind speed necessarily does not include any calm periods, whereas a 3-parameter distribution can do so. A major aim of this paper is to examine the extent to which this more complex model can improve estimates of wind energy potential.  Related to this aim is the most original part of the paper, namely to examine whether a new Bayesian method of estimating the parameters of a 2- or (particularly) a 3- parameter fitted distribution is good for this purpose. The paper achieves both these aims, and therefore warrants publication in some form.

Response: The authors would like to thank the reviewer for his encouraging comments as well as the suggestions and recommendations given to improve the manuscript.

Certainly sufficient detail is presented to enable a reader to make their own appraisal, including as to whether the  improvements are worthwhile in terms of practical estimation of wind energy potential, though the authors notably fail to discuss this aspect. Nevertheless, in my opinion, the main text of the paper includes some excessive detail that would benefit from moving into its Supplement or otherwise condensed.

Response: The authors have decided to delete some histograms from Section 8. Specifically, Figures 5, 8 and 10, since these histograms are apparently showing overlapping curves (to make the curves distinct, the histograms need to be enlarged considerably, which is not practical). We have also decided to delete eqns. 24 and 25 and the associated results and discussion as per the recommendation of Reviewer 2.

**SPECIFIC COMMENTS:**
Discussion and conclusions. A significant use of fitted Weibull distributions is to facilitate extrapolation from the measurement site to nearby sites using computer models like WAsP.  It is not self-evident that this can also be done with a 3-parameter fitted distribution, though the authors claim in a throw-away line in their conclusions that it can be done. Some such discussion should be included in the Discussion section, including a little on why such fitting can be useful the more so as it is bad practice to include a substantive point in a Conclusions section that is not raised earlier in the paper.

Response:  This discussion of possible incorporation of the 3-p Weibull distribution in software like WAsP is now included in the Discussion. It should be possible for software like WAsP to extrapolate to nearby sites, if all the details of the area of interest are available.

Table 8 is a valuable summary of results for average power production. Like Table 7, it supports the authors' qualitative conclusion that Bayesian-3P yields the best results (or nearly so) at all sites examined. MLE-2P represents the fit most widely used in the literature. It is surely worth noting that the Bayesian-3P (the method tested here) yields substantially lower relative error than does MLE-2P only at sites 3 and 7, which are the only sites with >2% of calms. In my opinion , it is very likely that for the other sites the difference of <2% between fitted and actual AEP  would  be overshadowed by the year-to-year changes in

actual AEP, so that the gain for engineering purposes of moving from MLE-2P to Bayesian-3P would barely be worthwhile. These fundamental points also warrant discussion, and perhaps an allusion in the Conclusions and the Abstract.

Response: The authors have now included the main points and the reviewer's suggestions in the Abstract, Discussion and Conclusions. The AEP is now replaced with Available Wind Power.
Note: Relative difference of Power from actual = relative difference of AEP from actual.

Literature cited or not cited. Much of the literature on wind energy makes broad statements about global trends in renewable energy, but perversely supported only by a citation to another narrowly focused paper on wind energy potential at a particular site. The first paragraph of the Introduction to this paper is a typical example. The assertion that many countries are turning to renewable energy is supported only a reference to Mostafacipour (2014), which is primarily yet another Weibull analysis of the wind energy distribution at a particular place; although that paper contains a similar sentence to the one in this paper, it too is supported only by a few references to similar single-site wind energy papers, and is any case rather dated. Surely a more appropriate basis for such general statements would be recent reports from one or more of the several institutions which specialize in energy statistics and their analysis, such as the International Energy Agency (IEA), the International Renewable Energy Agency (IRENA), the REN21 network with its annual Global Status Reports, The United Nations compilation of World Energy Statistics, or the BP Energy Statistical Review. Such references could also support the first sentence of the same paragraph.Conversely, there are a few general statements in the Introduction, which, while widely accepted, in an academic paper would normally warrant a reference. For example, that the Pacific Islands are among the most vulnerable places to climate change (line 45) could be readily referenced to the IPCC Sixth Assessment Report (Working Group 2) or similar.

Response: The authors appreciate the comment of the Reviewer to include references from international organizations like IRENA or IEA. The initial references are now replaced with references from these organizations; some more appropriate references will be added.

Site selection and measured data: For nearly a decade, the research group led by Professor Ahmed has been measuring the wind speed at sites across the Pacific Islands, using equipment like that described in sec. 2 of this paper. They have published directly measured data, usually for about 12 months, for sites in almost all the places listed in Table 1. The number of observations , helpfully given in Table 3, indicates that these measurements (for most of which no reference is made to the relevant paper) are usually the "actual' data used in this paper. . It is therefore surprising that the present paper draws on satellite data for sites 4 (Kadavu), 5 (Rarotonga), and 6 (Nuku'alofa). Why is this? What is the accuracy of that data, about which no methodological details are given. While the quality of satellite estimates of wind speed has no doubt improved, historically it tended to overestimate actual wind speeds (especially on hilly islands) by some 20% (Kumar & Prasad, Renewable Energy, 2010) .
Of course, the aim of the present paper is primarily mathematical , to examine how well various fitting methods match a set of "actual" wind speed data, rather than an engineering aim to examine the potential power available from the wind. Thus the accuracy of the underlying "actual" data does not matter for this mathematical purpose, particularly whether it systematically over- or under-estimates the wind speeds. A test of the curve-fitting method for sites with a greater proportion of calms than any of the sites examined so far would be of mathematical interest, even if such sites are unlikely to be of interest for production of electrical power. The most obvious examples in the Pacific for which a reasonable amount of data is available would be airports, which are usually sited in relatively calm sites for safety reasons! . Nadi airport for example has some 50 years of publicly available wind speed data at 3-hour intervals , of which >20% is calms (periods of 10 minutes with average wind speed <0.5 m/s) [FMS (2008), Surface winds at Nadi airport, Information Sheet No.38, , Nadi :Fiji Meteorological Service.]. It is important for wider use

that a tool for using the Bayesian 3-parameter method is available in the widely used statistical software package R , as pointed out  in sec.4.2.1.

Response:  1) For many sites, there is good match between measured and satellite data (e.g. Vanuatu and Tuvalu). The Kadavu site was chosen as a last resort, as there were land ownership issues. Even for other sites, due to land ownership issues, we could not install the measurement towers at the desired locations. 2) Table 3 now has references to our previous wind resource assessments. 3) The airport data are not reliable since the anemometers are not calibrated for many years.

**TECHNICAL COMMENTS:**

The paper often refers to its sites as being in the "equatorial" region, including in the Abstract and the Introduction. .  A more accurate  term here would be "tropical", as the term "equatorial" refers  more specifically to the zone from about 5degN to 5 degS , which includes the Intertropical Convergence Zone, which is the climatological region of the Doldrums, a calm region historically notorious among operators of sailing ships.

Response:  The term Equatorial is replaced with tropical; except for locations which are very close to the Equator.

The page length of the paper could be slightly condensed by merging the multiple similar figures into one or two appropriately labelled figures, each incorporating 4 or 6 of the charts. This would save repeating the information in the captions and legends. Incidentally, several of the charts appear to have only one curve plotted from among the 4 identified in the legend. If this is becaasue the curves co-incide (to visual accuracy) then the text should say so.

Response: The authors have decided to delete some histograms from Section 8. Specifically, Figures 5, 8 and 10, since these histograms are apparently showing overlapping curves (to make the curves distinct, the histograms need to be enlarged considerably, which is not practical). We have also decided to delete eqns. 24 and 25 and the associated results and discussion as per the recommendation of Reviewer 2. In addition to this, some of the Figures in the Appendix 2 will be deleted as per the suggestion from the reviewer.

Among  references cited, for which better sources are available include Mohanty (2012) [ which could be more appropriately replaced by the SPC's Framework for Energy Security and Resilience in the Pacific (FESRIP) 2021-2030 and/or its issues papers, or even Weir ("Renewable Energy in the Pacific Islands: its role and status, Renewable and Sustainable Energy Reviews, 2018) or Michalena et al ("Challenges for Pacific small island states", Energy Policy 2018) , which are slightly more recent than Mohanty's paper]. Also Kidmo (2015)  is focused on a far inland region of Cameroon and says nothing about developing country islands, notwithstanding line 53. The citations for MOE (2017) and PPA (2015) are appropriate, but the corresponding entries in the List of References are inadequate, since neither spells out the institution concerned, namely ( I guess) Ministry of Energy (Suva, Fiji) and Pacific Power Association , respectively

Response:  The authors appreciate the comment of the Reviewer to include references from international organizations like IRENA or IEA. The initial references are now replaced with references from these organizations; some more appropriate references will be added.

**Reviewer 2**

**GENERAL COMMENTS:**

This paper examines a Bayesian method, JAGS, for fitting probability distributions to wind data. JAGS is open source and approx. 20 years old, but it seems novel in the context of wind energy. Calculation of the posteriori probability distribution with JAGS requires MontCarlo simulation, and the authors used 10000 iterations for the examples in the paper. This does not appear computationally efficient compared to existing methods, but at least the results are similar to those of the well-known maximum-likelihood estimation method. The authors suggest a three-parameter Weibull distribution with a location parameter giving an offset of the wind-speed scale, as a better model than the traditional two-parameter Weibull distribution. Unfortunately, the fitted distributions for the six test cases in the paper are not convincing, since they all result in distributions with negative location parameter, i.e. the fitted distributions predicts finite probability of negative wind speeds. This is not physically possible. The fitted distributions are evaluated over the entire wind speed range. However, in the context of wind energy, it is more important to have an accurate model for wind speeds in the range of turbine operation. Wind resource estimation usually works with separate Weibull distributions for 12 or 16 wind-direction sectors and frequency of occurrence in each sector. This model is more accurate than a global Weibull distribution for wind speeds in all sectors.

Response: The authors would like to thank the reviewer for his detailed review and comments. The main concerns raised by the reviewer are responded to after some detailed analysis of his suggestions and/or recommendations. In the fitted three-parameter Weibull distribution, the location parameter was found negative and was found to capture the null wind speed (<= 1m/s) at the sites. Note that at all the sites, the low wind speeds are observed from 0.76% to 3.08% (see Table 6). In addition:

(i)    Sites with negative location parameter are also found in previous publications e.g. Wais (2017). He analyzed wind speed data at three sites in Poland and the location parameters were found as: - 1.2692170, - 0.2728700 and - 0.4318956.

(ii)   Although the location parameter is negative, the shape and scale parameters are so obtained that the wind speeds with the estimated parameters are positive with a finite probability shown in the following SAMPLE table, which presents the wind speed value at random p = 0.01, p = 0.05 and p=0.95. At a probability of less than 0.01, there are chances of negative wind speeds. So, the authors feel that negative location parameter is not a big issue.

| Site | $k$ | A | $\theta$ | U at 0.05p | U at 0.95p | U at 0.01p |
|------|-----|---|----------|------------|------------|------------|
| 1 | 2.777792 | 6.438856 | -0.380611 | 1.83 | 9.17 | 0.5784481 |
| 2 | 3.233794 | 7.532297 | -0.922046 | 2.082154 | 9.655944 | 0.8924875 |
| 3 | 2.636074 | 8.804806 | -1.546749 | 1.30673 | 11.78448 | 0.0007525313 |
| 4 | 2.401323 | 8.493057 | -0.042536 | 2.422822 | 13.37067 | |
| 5 | 2.569510 | 8.596043 | -0.323893 | 2.38755 | 12.85978 | |
| 6 | 2.522584 | 7.896873 | -0.047855 | 2.380784 | 12.16 | |
| 7 | 2.425388 | 7.807676 | -0.603869 | 2.250445 | 12.21713 | |

(iii)   Based on the comment of the reviewer, the location parameter was forced to remain positive (named here as 3-p Weibull+) and results were obtained. The updated **Table 5** shows that in Bayesian estimation, 3-p Weibull model fits better than the 2-p Weibull and 3-p Weibull+ (the model when the location parameter is forced to remain positive) for wind speed data at all the sites, except at Site 4 where 2-p Weibull fits (a special case of 3-p Weibull) better as the DIC is smaller.

• The table clearly shows that the model 3-p Weibull+ performs worse than the other two models.

**(Updated) Table 5: Estimated values of the parameters obtained using Bayesian technique and summary statistics.**

| Site | Distribution | Parameter | Mean | SD | 2.5% | 97.5% | DIC |
|------|--------------|-----------|------|-----|------|-------|-----|
| 1 | 2-p Weibull | k | 2.564487 | 0.008762 | 2.547292 | 2.581560 | 949864.6 |
|   |   | A | 6.019417 | 0.010876 | 5.998086 | 6.041268 |   |
|   | 3-p Weibull | k | 2.778195 | 0.018610 | 2.742305 | 2.814923 | **949587.6** |
|   |   | A | 6.439670 | 0.033898 | 6.373026 | 6.507357 |   |
|   |   | θ | -0.381144 | 0.029685 | -0.439400 | -0.323193 |   |
|   | 3-p Weibull+ | k | 2.563823 | 0.008847 | 2.546378 | 2.580583 | 949868.5 |
|   |   | A | 6.018284 | 0.010852 | 5.99728 | 6.039555 |   |
|   |   | θ | 0.001001 | 0.000991 | 2.72E-05 | 0.003657 |   |
| 2 | 2 parameter | k | 2.736076 | 0.008940 | 2.718410 | 2.753285 | 256112.2 |
|   |   | A | 6.535458 | 0.010427 | 6.515002 | 6.555774 |   |
|   | 3 parameter | k | 3.231943 | 0.023096 | 3.184961 | 3.273617 | **255252.4** |
|   |   | A | 7.528328 | 0.044253 | 7.438378 | 7.607035 |   |
|   |   | θ | -0.918508 | 0.040458 | -0.989749 | -0.835880 |   |
|   | 3-p Weibull+ | k | 2.73565 | 0.009057 | 2.717811 | 2.753553 | 256116.6 |
|   |   | A | 6.534764 | 0.010619 | 6.513872 | 6.555601 |   |
|   |   | θ | 0.000694 | 0.000701 | 1.81E-05 | 0.002556 |   |
| 3 | 2 parameter | k | 1.947640 | 0.005556 | 1.936812 | 1.958478 | 433669.2 |
|   |   | A | 7.017671 | 0.013022 | 6.992180 | 7.042895 |   |
|   | 3 parameter | k | 2.635850 | 0.016118 | 2.604492 | 2.667974 | **429666.8** |
|   |   | A | 8.804653 | 0.044092 | 8.718409 | 8.892835 |   |
|   |   | θ | -1.546999 | 0.038766 | -1.624995 | -1.472661 |   |
|   | 3-p Weibull+ | k | 1.947608 | 0.005361 | 1.93689 | 1.957928 | 433672.9 |
|   |   | A | 7.017143 | 0.012839 | 6.992167 | 7.042596 |   |
|   |   | θ | 0.0002 | 0.000198 | 5.36E-06 | 0.000734 |   |
| 4 | 2-p Weibull | k | 2.385596 | 0.019486 | 2.347390 | 2.424420 | **166458.7** |
|   |   | A | 8.446062 | 0.039889 | 8.367275 | 8.525195 |   |
|   | 3-p Weibull | k | 2.405809 | 0.026990 | 2.356128 | 2.461114 | 166459.8 |
|   |   | A | 8.507469 | 0.070222 | 8.377540 | 8.660753 |   |
|   |   | θ | -0.055714 | 0.051466 | -0.169674 | 0.031573 |   |
|   | 3-p Weibull+ | k | 2.378437 | 0.020051 | 2.339698 | 2.417936 | 166460.8 |
|   |   | A | 8.423957 | 0.043297 | 8.336452 | 8.508813 |   |
|   |   | θ | 0.018884 | 0.014624 | 0.000737 | 0.053911 |   |
| 5 | 2-p Weibull | k | 2.438528 | 0.011898 | 2.415595 | 2.462772 | 498423.7 |
|   |   | A | 8.233447 | 0.021645 | 8.190758 | 8.276122 |   |
|   | 3-p Weibull | k | 2.569184 | 0.024930 | 2.521912 | 2.619417 | **498375.**1 |
|   |   | A | 8.596434 | 0.065046 | 8.471752 | 8.729620 |   |
|   |   | θ | -0.324597 | 0.055076 | -0.438669 | -0.220035 |   |
|   | 3-p Weibull+ | k | 2.436463 | 0.011648 | 2.413392 | 2.459011 | 498427.1 |
|   |   | A | 8.229611 | 0.022021 | 8.185103 | 8.273141 |   |
|   |   | θ | 0.004002 | 0.003699 | 0.000103 | 0.013987 |   |
| 6 | 2-p Weibull | k | 2.502775 | 0.010741 | 2.482126 | 2.524369 | 603592.3 |
|   |   | A | 7.843795 | 0.018777 | 7.806656 | 7.880416 |   |
|   | 3-p Weibull | k | 2.523740 | 0.016243 | 2.492982 | 2.555752 | **603591.3** |
|   |   | A | 7.900740 | 0.038140 | 7.830501 | 7.978555 |   |
|   |   | θ | -0.051032 | 0.030016 | -0.113036 | 0.004279 |   |
|   | 3-p Weibull+ | k | 2.498836 | 0.011121 | 2.477168 | 2.52051 | 603595.1 |
|   |   | A | 7.832686 | 0.020837 | 7.791585 | 7.872396 |   |
|   |   | θ | 0.009855 | 0.008541 | 0.000325 | 0.031216 |   |
| 7 | 2-p Weibull | k | 2.153266 | 0.007378 | 2.139086 | 2.167660 | 1016627 |
|   |   | A | 7.111621 | 0.014636 | 7.083240 | 7.140584 |   |
|   | 3-p Weibull | k | 2.425984 | 0.016281 | 2.394208 | 2.457741 | **1016030** |

| | | | | | | |
|---|---|---|---|---|---|---|
| | A | 7.808854 | 0.040671 | 7.730066 | 7.892452 | |
| | θ | -0.604597 | 0.033817 | -0.673024 | -0.539979 | |
| 3-p Weibull+ | k | 2.153091 | 0.007379 | 2.139021 | 2.167982 | 1016631 |
| | A | 7.110982 | 0.01465 | 7.083335 | 7.139518 | |
| | θ | 0.000686 | 0.000672 | 1.80E-05 | 0.002457 | |

Furthermore, all the Goodness of fit measures were estimated for the four methods and the 3-p Weibull+ method. It can be seen from the updated Table 7 that the BAYESIAN.3P is the most efficient method for the sites 1, 2, 3, 5 and 7 as it gives the highest $R^2$ value and the least RMSE, MAE and MAPE values. This indicates that the 3-p Weibull with Bayesian estimates is a better method for wind energy assessment at these sites. For two sites (4 and 6), the BAYESIAN.3P+ performs better; however, the BAYESIAN.2P method also gives similar results, which is a special case of the proposed BAYESIAN.3P method.

**(Updated) Table 7: Goodness-of-fit measures for the four methods.**

| Site | Method | $R^2$ | COE | RMSE | MAE | MAPE |
|---|---|---|---|---|---|---|
| 1 | MLE.2P | 0.9982 | 0.9936 | 0.0983 | 0.0421 | 1.3052 |
| | MLE.3P | **0.9983** | 1.0011 | 0.0975 | **0.0239** | **0.8269** |
| | BAYESIAN.2P | 0.9982 | 0.9936 | 0.0983 | 0.0421 | 1.3053 |
| | BAYESIAN.3P | **0.9983** | 1.0011 | 0.0975 | **0.0239** | **0.8269** |
| | BAYESIAN.3P+ | 0.9982 | **0.9935** | **0.0943** | 0.0421 | 1.3051 |
| 2 | MLE.2P | 0.9949 | 1.0048 | 0.1655 | 0.1114 | 3.4259 |
| | MLE.3P | **0.9971** | **1.0034** | 0.1234 | 0.0661 | 1.7917 |
| | BAYESIAN.2P | 0.9949 | 1.0048 | 0.1655 | 0.1114 | 3.4259 |
| | BAYESIAN.3P | **0.9971** | 1.0041 | **0.1227** | **0.0645** | **1.7748** |
| | BAYESIAN.3P+ | 0.9949 | 1.0048 | 0.1655 | 0.1114 | 3.4259 |
| 3 | MLE.2P | 0.9696 | 0.9823 | 0.5675 | 0.4596 | 10.7507 |
| | MLE.3P | **0.9755** | 1.0681 | 0.5219 | 0.3029 | 7.9253 |
| | BAYESIAN.2P | 0.9696 | **0.9822** | 0.5677 | 0.4597 | 10.7516 |
| | BAYESIAN.3P | **0.9755** | 1.0679 | **0.5216** | **0.3027** | **7.9149** |
| | BAYESIAN.3P+ | 0.9697 | **0.9822** | 0.5676 | 0.4596 | 10.7512 |
| 4 | MLE.2P | 0.9787 | 1.0512 | 0.4983 | 0.1546 | 1.9778 |
| | MLE.3P | 0.9783 | 1.0523 | 0.5025 | 0.1522 | 1.9013 |
| | BAYESIAN.2P | 0.9787 | 1.0511 | 0.4981 | 0.1544 | 1.9724 |
| | BAYESIAN.3P | 0.9783 | 1.0524 | 0.5036 | **0.1518** | **1.8860** |
| | BAYESIAN.3P+ | **0.9789** | **1.0507** | **0.4962** | 0.1559 | 2.0176 |
| 5 | MLE.2P | 0.9983 | 1.0180 | 0.1371 | 0.1031 | 2.0079 |
| | MLE.3P | 0.9987 | 1.0186 | 0.1109 | 0.0787 | 1.5343 |
| | BAYESIAN.2P | 0.9983 | 1.0181 | 0.1372 | 0.1033 | 2.0074 |
| | BAYESIAN.3P | **0.9989** | 1.0183 | **0.1108** | **0.0785** | **1.5295** |
| | BAYESIAN.3P+ | 0.9983 | **1.0177** | 0.1375 | 0.1034 | 2.0129 |
| 6 | MLE.2P | 0.9911 | 1.0173 | 0.2809 | **0.1251** | 1.9635 |
| | MLE.3P | 0.9909 | 1.0181 | 0.2852 | 0.1257 | 1.9187 |
| | BAYESIAN.2P | 0.9911 | 1.0169 | 0.2806 | 0.1253 | 1.9673 |
| | BAYESIAN.3P | 0.9909 | 1.0180 | 0.2854 | 0.1258 | **1.9162** |
| | BAYESIAN.3P+ | **0.9912** | **1.0168** | **0.2798** | **0.1251** | 1.9771 |
| 7 | MLE.2P | 0.9973 | **0.9991** | 0.1607 | 0.1126 | 3.4174 |
| | MLE.3P | 0.9987 | 1.0152 | 0.1069 | 0.0718 | 2.0482 |
| | BAYESIAN.2P | 0.9973 | 0.9992 | 0.1609 | 0.1130 | 3.4244 |
| | BAYESIAN.3P | **0.9988** | 1.0152 | **0.1068** | **0.0717** | **2.0476** |
| | BAYESIAN.3P+ | 0.9973 | 0.9992 | 0.1609 | 0.1130 | 3.4247 |

Concerning the comment on fitting only the wind speeds in the operating range of the turbine (≥3.5 m/s), please note that this paper is only estimating the available power. Also, different turbines have different cut-in speeds; nowadays, they are even designing turbines with cut-in speeds of 2 m/s. Hence, it is difficult to fix a wind speed to remove lower wind speed data. However, to respond to the reviewer, we fitted the Weibull curves with wind speeds of 3.5 m/s and above for site 1 and estimated the different goodness of fit values. The results are shown below, which demonstrate that the Weibull curve gets distorted and the goodness of fit values indicate higher errors.

[Figure]

Goodness of Fit:

|  | Rsq | COE | RMSE | MAE | MAPE |
|---|---|---|---|---|---|
| MLE.2P | 0.961936 | 1.118722 | 0.372791 | 0.207599 | 3.482373 |
| MLE.3P | 0.995556 | 0.969421 | 0.120654 | 0.084971 | 1.322561 |
| BAYESIAN.2P | 0.961948 | 1.118627 | 0.372706 | 0.207574 | 3.482144 |
| BAYESIAN.3P | 0.995514 | 0.968467 | 0.121347 | 0.085516 | 1.328605 |

**SPECIFIC COMMENTS**
Equation 4 should only allow positive values of the location parameter, and the distribution fitting method should enforce this restriction.

**Response:** For the tropical region, where the percentage of null wind speeds is significant at most of the sites, if the location parameter in eqn. (4) is forced to be positive:

(i)     the 3-parameter Weibull will not remain applicable. In other words, if $θ > 0$, it is likely that $θ = 0$, which reduces to 2-parameter Weibull. The results contradict with Wais (2017) "*The three-parameter Weibull distribution takes into account the frequency of null winds and it can represent a useful alternative to the two-parameter Weibull distribution for the wind with considerable null wind probability and a higher fraction of lower speed winds*".

(ii)    As the MLE estimate gives the location parameter of 3-parameter Weibull negative, MLE estimation is inappropriate (Green et al. 1994). Thus, only Bayesian estimate is applicable to the wind speed data at such locations.

Further, for our experimental data, it is found that the forecasted wind speed is rarely (even less than 1%) negative if the location parameter is allowed to be negative. It can be seen from the updated Table 8 below that the BAYESIAN.3P model gives better results than the one with forced positive location parameter (it is overestimating the power).

**(Updated) Table 8: Estimated power and relative error in power estimation.**

| Site | Methods | Power (kW) | RE (%) |
|---|---|---|---|
| 1 | Actual | 108457 | 0.00 |

|   |            |        |       |
|---|------------|--------|-------|
|   | MLE.2P     | 110306 | 1.68  |
|   | MLE.3P     | 110061 | 1.46  |
|   | BAYESIAN.2P | 110298 | 1.67 |
|   | **BAYESIAN.3P** | **110051** | **1.45** |
|   | BAYESIAN.3P+ | 110300 | 1.67 |
|   | Actual     | 133525 | 0.00  |
|   | MLE.2P     | 136039 | 1.85  |
| 2 | MLE.3P     | 135765 | 1.65  |
|   | BAYESIAN.2P | 136030 | 1.84 |
|   | **BAYESIAN.3P** | **135753** | **1.64** |
|   | BAYESIAN.3P+ | 136036 | 1.85 |
|   | Actual     | 206184 | 0.00  |
|   | MLE.2P     | 220464 | 6.48  |
| 3 | MLE.3P     | 209474 | 1.57  |
|   | BAYESIAN.2P | 220556 | 6.52 |
|   | **BAYESIAN.3P** | **209458** | **1.56** |
|   | BAYESIAN.3P+ | 220525 | 6.50 |
|   | Actual     | 322492 | 0.00  |
|   | MLE.2P     | 319732 | -0.86 |
| 4 | MLE.3P     | 319627 | -0.90 |
|   | **BAYESIAN.2P** | **319812** | **-0.84** |
|   | BAYESIAN.3P | 319591 | -0.91 |
|   | BAYESIAN.3P+ | 319775 | -0.85 |
|   | Actual     | 290717 | 0.00  |
|   | MLE.2P     | 291809 | 0.37  |
| 5 | MLE.3P     | 290928 | 0.07  |
|   | BAYESIAN.2P | 291696 | 0.34 |
|   | **BAYESIAN.3P** | **290889** | **0.06** |
|   | BAYESIAN.3P+ | 291815 | 0.38 |
|   | Actual     | 249192 | 0.00  |
|   | MLE.2P     | 247792 | -0.56 |
| 6 | MLE.3P     | 247770 | -0.57 |
|   | **BAYESIAN.2P** | **247848** | **-0.54** |
|   | BAYESIAN.3P | 247808 | -0.56 |
|   | BAYESIAN.3P+ | 247840 | -0.55 |
|   | Actual     | 204938 | 0.00  |
|   | MLE.2P     | 207543 | 1.26  |
| 7 | MLE.3P     | 204781 | -0.08 |
|   | BAYESIAN.2P | 207487 | 1.23 |
|   | **BAYESIAN.3P** | **204790** | **-0.07** |
|   | BAYESIAN.3P+ | 207493 | 1.23 |

Equations 21-25 does not include a wind-turbine power curve, so they are not predictions of annual energy production. I suggest that you avoid multiplication by the rotor-swept area in equations 22-23, and refer to the result as 'wind power density', which is a standard term in wind energy. Furthermore, you can delete equations 24-25.

Response: The authors have decided to delete Equations 24-25 and the annual energy production. However, eqns. 22-23 are retained and the power is now reported as total available wind power similar to the previous work discussed.

It seems like the last column (AEP) in table 8 is calculated by equations 24-25. AEP normally refers to the 'annual energy production' for a specific project with a selected turbine, and you need the turbine power curve to estimate this. I suggest that you delete this column or substitute by wind power density.

Response: AEP calculations are now removed and only total available wind power estimations are reported.

The integration range in equations 22- 23 covers all positive wind speeds, which is natural from a physical point of view. However, the fitted 3-parameter distributions all have negative location parameters, so you are not integrating over the entire probability space. In one case, you only integrate over 98.5% of the probability, which gives the result a negative bias. The conclusion is that we cannot apply the three-parameter Weibull distribution with negative location parameter for a wind speed distribution.

Response: From the updated tables 5, 7 and 8, it is clear that the three-parameter Weibull distribution with negative location parameter is the most appropriate model.
Note: If we integrate, we can use up to 99.09% of probability space (see table above).

Figures 5-11 compares fitted distributions with bin statistics of wind speeds at six met masts. It seems unnecessary to include this many similar plots, especially since tables 4 and 5 summarize the key statistics.

Response: The authors have decided to delete Figs 5, 8 and 10, since it is also the recommendation of reviewer 1.

**TECHNICAL CORRECTIONS**
The text should explain why some numbers in tables 4, 5 and 7 are printed with bold.

Response: The values were highlighted to show that the 3P method is performing better.

**References:**

Wais, P.: Two and three-parameter Weibull distribution in available wind power analysis, Renewable energy, 103, 15-29, doi: 10.1016/j.renene.2016.10.041, 2017.

Green, E.J., Roesch, F.A. Jr., Smith, A.F.M. and Strawderman, W.E.: Bayesian Estimation for the Three-Parameter Weibull Distribution with Tree Diameter Data, Biometrics, 50, 254-269, https://doi.org/10.2307/2533217, 1994.

---

## Referee Report (RR1)

**Review of WES-2022-70**

**Title** Bayesian method for estimating Weibull parameters for wind resource assessment in the Equatorial region: A comparison between two-parameter and three-parameter Weibull distributions

**Authors** M. Golam Mostafa Khan and M. Rafiuddin Ahmed

**Iteration** Revised submission

**Specific comments**

- Equation 4 is now restricted to positive wind speeds, as it should. However, the integral of this will be less than one when the offshore parameter is negative, so it is not a proper probability distribution. You can repair this by adding a finite probability of calm wind speed, see below.

- I am happy about the changes to equation 21-25 and table 8. However, it would be better to substitute the term 'available power' by 'wind power density', which is standard in wind energy.

- Figures 1-3 shows the variations of repeated Bayesian estimators. This information is necessary for the reader.

- Figures 5-11 compares fitted distributions with bin statistics of wind speeds at six met masts. It seems unnecessary to include this many similar plots, especially since tables 4 and 5 summarises key statistics.

- It would be relevant to compare the times of computation by the Maximum likelihood method and the Bayesian fitting by JAGS. The results are very similar, but what about the efficiency?

**Technical comments**

- The manuscript should explain why some numbers in tables 4, 5 and 7 are printed with bold.

**Note about Equation 4**

For **positive offset parameter** $(\theta \geq 0)$ we can use the standard three-parameter Weibull distribution.

$$p(u\,|k, A, \theta) = \begin{cases} \frac{k}{A}\left(\frac{u-\theta}{A}\right)^{k-1} \exp\left\{-\left(\frac{u-\theta}{A}\right)^k\right\} & \text{for } u \geq \theta \\ 0 & \text{for } u < \theta \end{cases}$$

For **negative offset parameter** $(\theta < 0)$ we have a problem, since then the integral of the standard probability distribution over non-negative wind speeds becomes less than one. Instead, you could combine the truncated version the 3-parameter Weibull distribution supporting positive wind speeds only with a finite probability for calm situations $u = 0$, e.g. written like

$$p(u\,|k, A, \theta, p_{calm}) = \begin{cases} p_{calm} \cdot \delta(u) + (1 - p_{calm}) \frac{k}{A}\left(\frac{u-\theta}{A}\right)^{k-1} \frac{\exp\left\{-\left(\frac{u-\theta}{A}\right)^k\right\}}{\exp\left\{-\left(\frac{-\theta}{A}\right)^k\right\}} & \text{for } u \geq 0 \\ 0 & \text{for } u < 0 \end{cases}$$

using Kronecker's delta function $\delta(u)$. You could simplify this mixed probability distribution to an expression with three parameters, if require postulate the probability of calm situations to be equal to the clipped-off part of the Weibull distribution. This would change the expression to

$$p(u\,|k, A, \theta) = \begin{cases} p_{calm} \cdot \delta(u) + \frac{k}{A}\left(\frac{u-\theta}{A}\right)^{k-1} \exp\left\{-\left(\frac{u-\theta}{A}\right)^k\right\} & \text{for } u \geq 0 \\ 0 & \text{for } u < 0 \end{cases}$$

with the somewhat arbitrary **constraint**

$$p_{calm} \overset{\text{def}}{=} 1 - \exp\left\{-\left(\frac{-\theta}{A}\right)^k\right\}$$

Without this constraint, the probability model would have four parameters – $k$, $A$, $\theta$ and $p_{calm}$.

---

## Author Response (AR2)

**Response to Reviewers' Comments (WES-2022-70)**

**Specific comments**

• Equation 4 is now restricted to positive wind speeds, as it should. However, the integral of this will be less than one when the offshore parameter is negative, so it is not a proper probability distribution. You can repair this by adding a finite probability of calm wind speed, see below.

**Response:** The authors would like to thank the reviewer for working out and recommending a proper probability distribution to encounter the situation when the shift parameter is negative, which can be used when a significant loss in the integral value. The authors have incorporated the probability distribution as suggested (please see Equations 6-7) in the revised manuscript as a solution to those researchers who get the integral value considerably less than 1. In this paper, although the shift parameter is negative for all the sites, all the integral values are very close to 1, which is supposed to be the actual value, and the percentage deviation from the actual value is only 0-1.7% in the MLE estimate, and 0-1% in the Bayesian estimate, which is insignificant as shown below:

**Table: Integral values of 3-parameter Weibull with MLE Estimate**

| Site | $k$ | $A$ | $\theta$ | Integral value ($x \geq$ 0) | Deviation from actual value (%) |
|------|-----------|----------|-----------|---------|-----|
| 1 | 2.777792 | 6.438856 | -0.380611 | 0.99961 | 0.0 |
| 2 | 3.233794 | 7.532297 | -0.922046 | 0.99887 | 0.1 |
| 3 | 2.636074 | 8.804806 | -1.546749 | 0.98984 | 1.0 |
| 4 | 2.401323 | 8.493057 | -0.042536 | 0.98339 | 1.7 |
| 5 | 2.569510 | 8.596043 | -0.323893 | 0.99978 | 0.0 |
| 6 | 2.522584 | 7.896873 | -0.047855 | 0.99999 | 0.0 |
| 7 | 2.425388 | 7.807676 | -0.603869 | 0.99799 | 0.2 |

**Table: Integral values of 3-parameter Weibull with Bayesian Estimate**

| Site | $k$ | $A$ | $\theta$ | Integral value ($x \geq 0$) | Deviation from actual value (%) |
|------|-----------|----------|-----------|---------|-----|
| 1 | 2.778195 | 6.43967 | -0.381144 | 0.99961 | 0.0 |
| 2 | 3.231943 | 7.528328 | -0.918508 | 0.99888 | 0.1 |
| 3 | 2.63585 | 8.804653 | -1.546999 | 0.98983 | 1.0 |
| 4 | 2.405809 | 8.507469 | -0.055714 | 0.99999 | 1.7 |
| 5 | 2.569184 | 8.596434 | -0.324597 | 0.99977 | 0.0 |
| 6 | 2.52374 | 7.90074 | -0.051032 | 0.99999 | 0.0 |
| 7 | 2.425984 | 7.808854 | -0.604597 | 0.99799 | 0.2 |

• I am happy about the changes to equation 21-25 and table 8. However, it would be better to substitute the term 'available power' by 'wind power density', which is standard in wind energy.

**Response:** 'available power' has been substituted by 'wind power density' as suggested.

• Figures 1-3 shows the variations of repeated Bayesian estimators. This information is necessary for the reader.

**Response:** This information is now added in the revised manuscript in the Results section.

• Figures 5-11 compares fitted distributions with bin statistics of wind speeds at six met masts. It seems unnecessary to include this many similar plots, especially since tables 4 and 5 summarises key statistics.

**Response:** Based on the above suggestion, only four fitted distributions are retained and three are deleted.

• It would be relevant to compare the times of computation by the Maximum likelihood method and the Bayesian fitting by JAGS. The results are very similar, but what about the efficiency?

**Response:** The computation times taken for the 2-p and 3-p methods for site 1 are shown in the table below. However, a direct comparison of the computational time would not be appropriate as the Bayesian method is a simulation based technique which take longer time than the standard software based MLE technique. A statement regarding this now added in the revised manuscript.

|  | Computation Time | |
|---|---|---|
|  | MLE | Bayesian |
| 2P- Weibull | approx 1 min | approx. 24 min |
| 3P- Weibull | approx. 2 min | Approx. 34 min |

**Technical comments**
• The manuscript should explain why some numbers in tables 4, 5 and 7 are printed with bold.

**Response:** The values that are highlighted are the best goodness of fit/error estimates. It can be seen from the highlighted values that the 3P method is performing better.

---

## Author Response (AR3)

**Response to Associate Editor Comments**

Congratulations! You have sufficiently addressed all the questions and suggestions requested by the reviewer. Your manuscript is now very close to being accepted for publication in WES.

However, a few technical issues still need to be corrected. Please carefully read the instructions for manuscripts, especially concerning SI units, tables and figures format and references. The instructions are available from https://www.wind-energy-science.net/submission.html.

Response: The authors would like to thank the Associate Editor for accepting the revised manuscript and for pointing out the issues that need to be fixed. The authors have now made the necessary changes including ensuring correct units, formatting and providing descriptive captions for figures and tables as well as references.